# A 20-year (1998–2017) global sea surface dimethyl sulfide gridded dataset with daily resolution

**Shengqian Zhou**[1], **Ying Chen**[1,2,3], **Shan Huang**[4,5], **Xianda Gong**[6,7], **Guipeng Yang**[8,9,10], **Honghai Zhang**[8,9,10], **Hartmut Herrmann**[5], **Alfred Wiedensohler**[5], **Laurent Poulain**[5], **Yan Zhang**[1,2], **Fanghui Wang**[1], **Zongjun Xu**[1], and **Ke Yan**[1]

[1]Shanghai Key Laboratory of Atmospheric Particle Pollution Prevention, Department of Environmental Science & Engineering, Fudan University, 200438 Shanghai, China
[2]Institute of Eco-Chongming (IEC), National Observations and Research Station for Wetland Ecosystems of the Yangtze Estuary, 200062 Shanghai, China
[3]Institute of Atmospheric Sciences, Fudan University, 200438 Shanghai, China
[4]Institute for Environmental and Climate Research, Jinan University, 511443 Guangzhou, China
[5]Atmospheric Chemistry Department, Leibniz Institute for Tropospheric Research, 04318 Leipzig, Germany
[6]Research Center for Industries of the Future, Westlake University, 310030 Hangzhou, China
[7]Key Laboratory of Coastal Environment and Resources of Zhejiang Province, School of Engineering, Westlake University, 310030 Hangzhou, China
[8]Frontiers Science Center for Deep Ocean Multispheres and Earth System and Key Laboratory of Marine Chemistry Theory and Technology, Ministry of Education, Ocean University of China, 266100 Qingdao, China
[9]Laboratory for Marine Ecology and Environmental Science, Qingdao National Laboratory for Marine Science and Technology, 266071 Qingdao, China
[10]College of Chemistry and Chemical Engineering, Ocean University of China, 266100 Qingdao, China

**Correspondence:** Ying Chen (yingchen@fudan.edu.cn)

**Abstract.** The oceanic emission of dimethyl sulfide (DMS) plays a vital role in the Earth's climate system and constitutes a substantial source of uncertainty when evaluating aerosol radiative forcing. Currently, the widely used monthly climatology of sea surface DMS concentration falls short of meeting the requirement for accurately simulating DMS-derived aerosols with chemical transport models. Hence, there is an urgent need for a high-resolution, multi-year global sea surface DMS dataset. Here we develop an artificial neural network ensemble model that uses nine environmental factors as input features and captures the variability of the DMS concentration across different oceanic regions well. Subsequently, a global sea surface DMS concentration and flux dataset ($1° \times 1°$) with daily resolution spanning from 1998 to 2017 is established. According to this dataset, the global annual average concentration was $\sim 1.71$ nM, and the annual total emissions were $\sim 17.2$ Tg S yr$^{-1}$, with $\sim 60\%$ originating from the Southern Hemisphere. While overall seasonal variations are consistent with previous DMS climatologies, notable differences exist in regional-scale spatial distributions. The new dataset enables further investigations into daily and decadal variations. Throughout the period 1998–2017, the global annual average concentration exhibited a slight decrease, while total emissions showed no significant trend. The DMS flux from our dataset showed a stronger correlation with the observed atmospheric methanesulfonic acid concentration compared to those from previous monthly climatologies. Therefore, it can serve as an improved emission inventory of oceanic DMS and has the potential to enhance the simulation of DMS-derived aerosols and associated radiative effects. The new DMS gridded products are available at https://doi.org/10.5281/zenodo.11879900 (Zhou et al., 2024).

## 1 Introduction

Dimethyl sulfide (DMS), primarily produced by ocean biota, accounts for more than half of natural sulfur emissions and significantly contributes to the sulfur dioxide in the troposphere (Sheng et al., 2015; Andreae, 1990), which can be oxidized to sulfuric acid and form sulfate aerosols (Barnes et al., 2006; Hoffmann et al., 2016). Sulfate aerosols play an important role in climate systems by scattering solar radiation, changing the cloud condensation nuclei (CCN) population, and altering cloud properties (Masson-Delmotte et al., 2021). Recent studies have proven that CCN over the remote ocean and polar regions are primarily composed of non-sea-salt sulfate (nss-$SO_4^{2-}$) (Quinn et al., 2017; Park et al., 2021). Given the weak influence of anthropogenic $SO_2$ over open oceans, marine biogenic DMS emerges as a crucial source of nss-$SO_4^{2-}$, thus regulating oceanic climate (McCoy et al., 2015). Accordingly, DMS has been suggested to be the key substance in the postulated feedback loop of marine phytoplankton to climate warming (the "CLAW" hypothesis) (Charlson et al., 1987), although this is the subject of several controversies (Quinn and Bates, 2011). To accurately simulate the climate effects of DMS-derived aerosols, high-fidelity and high-resolution data on sea surface DMS concentrations and emission fluxes are required, along with further explorations of complex atmospheric chemical and physical processes (Hoffmann et al., 2016; Novak et al., 2021). It has been indicated that the uncertainty in DMS emission flux is the second-largest contributor to the overall uncertainty associated with natural aerosols when evaluating the aerosol indirect radiative forcing (Carslaw et al., 2013). Therefore, understanding the spatiotemporal variations of DMS in global oceans is currently an important task.

There are complex production and consumption mechanisms of DMS in the upper ocean, which makes it difficult to capture the dynamics and distributions of sea surface DMS across different regions well. Dimethylsulfoniopropionate (DMSP), the major precursor of DMS, is synthesized mainly by phytoplankton in the photic zone and has a variety of physiological functions in algal cells (Stefels, 2000; Sunda et al., 2002; McParland and Levine, 2018). The DMSP yield varies significantly among algal species (Stefels et al., 2007; Keller et al., 1989), and DMS can be produced through DMSP intracellular and extracellular cleavage by both algae and bacteria (Alcolombri et al., 2015; Zhang et al., 2019). Therefore, the oceanic DMS produced via multiple pathways can be affected by many biotic and abiotic factors, such as temperature, salinity, solar radiation, mixed-layer depth, nutrients, oxygen, and acidity (Simó and Pedrós-Alió, 1999a; Vallina and Simó, 2007; Stefels, 2000; Zindler et al., 2014; Six et al., 2013; Omori et al., 2015; Stefels et al., 2007). In addition, seawater DMS undergoes various removal pathways (bacterial consumption, photodegradation,

sea-to-air ventilation, etc.), further complicating its cycling (Stefels et al., 2007; Galí and Simó, 2015; Hopkins et al., 2023). Therefore, although previous studies have developed several empirical algorithms (Simó and Dachs, 2002; Belviso et al., 2004b; Vallina and Simó, 2007) and process-embedded prognostic models (Kloster et al., 2006; Vogt et al., 2010; Belviso et al., 2011; Wang et al., 2015) based on relevant variables (mixed-layer depth, chlorophyll $a$, nutrients, radiation, phytoplankton group, etc.) to estimate the distribution of DMS, their results showed significantly different patterns and inconsistency with observations in many regions (Tesdal et al., 2016; Belviso et al., 2004a). Recently, Galí et al. (2018) developed a new empirical algorithm based on a parameterization of DMSP (Galí et al., 2015). The estimated DMS field exhibited a generally higher consistency with observations than those derived from the previous algorithms SD02 (Simó and Dachs, 2002) and VS07 (Vallina and Simó, 2007), but this method did not consider the influences of nutrients and still exhibited substantial biases in certain regions (e.g., near the Antarctic).

Since Lovelock et al. (1972) first discovered the ubiquitous presence of DMS in seawater, numerous observations of sea surface DMS have been conducted worldwide, yielding a substantial volume of observational data to date. Based on these worldwide measurements, a monthly climatology of global DMS can be generated through interpolation and extrapolation (Hulswar et al., 2022; Kettle et al., 1999; Lana et al., 2011). The latest version incorporated 873 539 raw observations (48 898 after data filtration and unification for climatology development), and the estimated global annual mean concentration and total flux are 2.26 nM and 27.1 Tg S yr$^{-1}$, respectively (Hulswar et al., 2022). However, despite the abundance of data, significant spatial and temporal disparities persist, potentially introducing large uncertainties into regions or periods with sparse observations. Furthermore, the observational data from the same month in different years were combined for interpolation and extrapolation, and interannual variations cannot be investigated by this approach.

In recent years, the application of data-driven approaches like machine learning to Earth system science has drawn more and more attention. Compared with traditional approaches, machine learning explores a larger function space and captures more hidden information from big data; hence, it often provides better prediction performance (Reichstein et al., 2019; Zheng et al., 2020; Bergen et al., 2019). For instance, a recent study demonstrated that an artificial neural network (ANN) can capture much more ($\sim 66\%$) of the raw data variance than multilinear regression ($\sim 39\%$), and a global monthly climatology of sea surface DMS concentration has been developed based on the ANN model (Wang et al., 2020). Machine learning techniques have also been used to simulate the distribution of DMS in the Arctic (Humphries et al., 2012; Qu et al., 2016), North Atlantic Ocean (Bell et

al., 2021; Mansour et al., 2023), northeastern Pacific Ocean (McNabb and Tortell, 2022), Southern Ocean (McNabb and Tortell, 2023), and East Asia (Zhao et al., 2022).

However, to our best knowledge, there is currently no global-scale sea surface gridded DMS dataset with both high time resolution (daily) and long-term coverage (>10 years). Such a dataset is urgently needed for modeling the atmospheric processes and climatic implications of oceanic DMS. The sea surface concentration and sea-to-air emission flux of DMS can vary greatly from day to day (Simó and Pedrós-Alió, 1999b), and the emitted DMS exerts effects on the atmosphere over timescales of several hours to days. Relying solely on a monthly climatology of DMS as the emission inventory may result in a failure to capture important details and could lead to large modeling biases compared to observed concentrations of atmospheric DMS or its oxidation products (Chen et al., 2018; Fung et al., 2022).

Here, we build a 20-year (1998–2017) global sea surface DMS gridded dataset ($1° \times 1°$) with daily resolution based on a data-driven machine learning approach (ANN ensemble). This product can improve our understanding of the spatiotemporal variations of oceanic DMS. More importantly, it can serve as an updated emission inventory of marine biogenic DMS for chemical transport models, which is beneficial for enhancing the simulation of atmospheric processes of DMS and reducing the uncertainties in marine aerosol's climate effects. This paper consists of four main parts, as depicted in Fig. 1: (1) the development of the machine learning model based on global DMS measurements and nine ancillary environmental variables, (2) the derived spatial and temporal distributions of DMS and comparisons with previous estimates, (3) an example showing the superiority of our newly developed DMS field through its correlation with atmospheric biogenic sulfur, and (4) the uncertainties and limitations inherent in our approach and the resulting data product.

## 2 Methodology

### 2.1 Input datasets

The in situ DMS measurement data used for training the machine learning model were primarily sourced from the Global Surface Seawater DMS (GSSD) database (Kettle et al., 1999). The GSSD database contains a total of 87 801 DMS measurements collected across 266 cruise and fixed-site observation campaigns from 11 March 1972 to 27 August 2017 (https://saga.pmel.noaa.gov/dms/, last access: 1 April 2020). Hulswar et al. (2022) consolidated other DMS measurements not included in the GSSD database to establish an updated DMS climatology. Here, we incorporated these additional data predating 2017, which originate from eight campaigns (number of samples 6711). The spatial distribution of these 94 512 in situ observational data values in total is shown in Fig. S1 in the Supplement, which covers all major regions of the global ocean.

We selected nine environmental variables relevant to DMS biogeochemical processes as input features, including chlorophyll $a$ (Chl $a$), sea surface temperature (SST), mixed-layer depth (MLD), nitrate, phosphate, silicate, dissolved oxygen (DO), downward shortwave radiation flux (DSWF), and sea surface salinity (SSS). The data sources and relevant information for these nine input variables and DMS are listed in Table 1. Chl $a$ data were obtained from both in situ observations co-located with DMS data and satellite remote sensing products (Copernicus-GlobColour, Level 4; daily; $0.042° \times 0.042°$). The dataset Copernicus-GlobColour, Level 4, integrates multiple upstream sensors, including SeaWiFS, MODIS Aqua and Terra, MERIS, VIIRS-SNPP and JPSS1, and OLCI-S3A and OLCI-S3B, and an interpolation procedure is applied to fill in missing data (Garnesson et al., 2019). Daily SST data ($0.25° \times 0.25°$) were obtained from the NOAA OI SST V2 high-resolution blended reanalysis dataset (Huang et al., 2021). Daily MLD, DSWF, and SSS data were obtained from version 4 release 4 (V4r4) of the modeling outputs of NASA's Estimating the Circulation and Climate of the Ocean (ECCO) consortium (Forget et al., 2015). The sea surface concentrations of nitrate, phosphate, silicate, and DO were obtained from the CMEMS global biogeochemical multi-year hindcast dataset (daily; $0.25° \times 0.25°$). Surface wind speed (WS) and sea ice fraction (SI) data are needed in the calculation of the sea-to-air flux (details are provided in Sect. 2.4.2). Here, we utilized the daily 10 m WS data from ECCO V4r4 and the daily SI data from NOAA OI SST V2. Since there were multiple different spatial grids among all the datasets, the data were matched up as described in the next section.

### 2.2 Data preprocessing for model development

The data extraction and matchup were performed based on the sampling location and time associated with each DMS measurement record as well as the temporal range and grid distribution of each variable. For satellite-retrieved Chl $a$, the data for the grids covering DMS sampling locations were extracted. If the data for the corresponding grid were missing, the average value for the $5 \times 5$ grids nearby was calculated and used. For other variables, only values in grids matching DMS sampling locations were extracted.

There are in situ Chl $a$ measurements that are co-located with certain GSSD data. They were also used along with the satellite-retrieved Chl $a$. In situ Chl $a$ measurements with low precision (defined as <0.1 mg m$^{-3}$ when the number of significant digits is 1) were removed. For a specific in situ observation campaign, if the number of low-precision values was larger than 10 and accounted for more than half of the values, all in situ Chl $a$ data from that campaign were excluded. In addition, the in situ Chl $a$ data in the GSSD database were measured by two different methods: Turner fluorome-

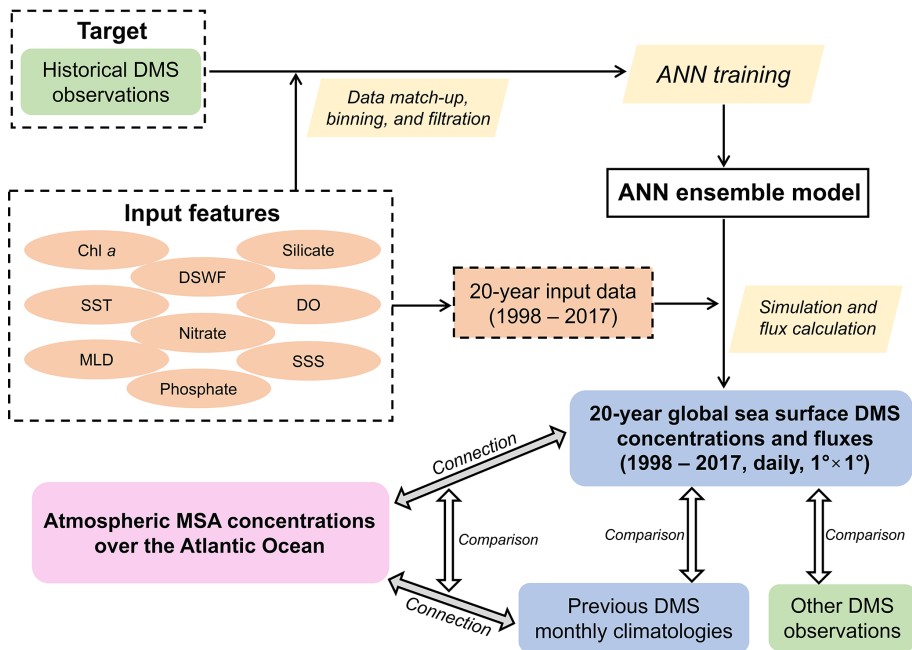

**Figure 1.** Flowchart of this study, including the development of the ANN ensemble model, the construction of the new DMS gridded dataset, and subsequent evaluations of this product.

try and high-performance liquid chromatography (HPLC). In order to improve mutual consistency, a conversion between the data from these two methods was applied, and then the in situ Chl *a* concentrations were adjusted to match them up with the satellite Chl *a* using the functions described in Galí et al. (2015). After that, the statistical outliers among all the $\log_{10}$(Chl *a*) values (i.e., those outside a range defined as the average $\pm 3$ standard deviations) were eliminated. A comparison between the in situ and satellite-retrieved Chl *a* data is shown in Fig. S2. The strong consistency between in situ and daily satellite Chl *a* data ($R^2 > 0.5$; RMSE $< 0.4$) provides the rationale for integrating these datasets. The $\log_{10}$ transformation was applied to make the data distribution close to a normal distribution. When finally selecting the $\log_{10}$(Chl *a*) value corresponding to each DMS data value, in situ data were prioritized where available; otherwise, the satellite-retrieved data were used.

The DMS values and extracted values of MLD and three nutrients (nitrate, phosphate, and silicate) were also $\log_{10}$-transformed. The statistical outliers for each variable were excluded as mentioned above. After data filtration, a total of 633 361 CE1 samples with valid data for all variables were obtained. To avoid a data aggregation bias stemming from the clustering of multiple data points within a narrow temporal range and spatial range (i.e., obtained on the same day and within a region smaller than $0.05° \times 0.05°$), these data points were averaged. Consequently, 41 157 binned samples were utilized for subsequent model development; their spatial distribution is depicted in Fig. 2a.

We divided the global ocean into nine regions based on Longhurst's biomes (Longhurst, 1998). There are six biomes in Longhurst's definition, including Coastal, Polar_N, Polar_S, Westerlies_N, Westerlies_S, and Trades (the .shp file of Longhurst's biomes and provinces was downloaded from https://www.marineregions.org/downloads.php#longhurst, last access: 16 April 2020). We further divided Westerlies_N into Westerlies_N_Pacific and Westerlies_N_Atlantic and divided Trades into Trades_ Pacific, Trades_Indian, and Trades_Atlantic based on the different oceanic basins, as shown in Fig. 2b. It is noteworthy that there are 11 237 samples in the Coastal region, constituting 27.3 % of the entire sample set, despite the Coastal biome accounting for only 9.7 % of the global ocean area. Given the distinct physiochemical and biological conditions of seawater in coastal seas compared to other regions, the disproportionately higher density of samples within the Coastal biome might cause the model to overly prioritize this region. To mitigate this data imbalance and ensure the model captures broader patterns in open oceans, we adjusted the data distribution during the model training and validation processes. Specifically, for each training session, a portion of coastal samples was randomly removed to ensure that the proportion of coastal samples in the total sample set (denoted as $F_{coastal}$) matched the coastal proportion of the total area.

## 2.3  Artificial neural network training and validation

The 41 157 binned samples obtained after the previously mentioned data preprocessing were used to develop the ar-

**Table 1.** The data sources and relevant information for the variables used for model development, DMS simulation, and flux calculation.

| Variable | Data source | URL | Temporal resolution | Temporal coverage | Spatial grid |
|---|---|---|---|---|---|
| DMS | GSSD database | https://saga.pmel.noaa.gov/dms/ (last access: 1 April 2020) | In situ | March 1972–August 2017 | – |
| | Other campaigns included in Hulswar et al. (2022) | https://data.mendeley.com/datasets/ hyn62spny2/1 (last access: 25 November 2023) | In situ | February 2000–June 2016 | – |
| Chl $a$ | GSSD database | https://saga.pmel.noaa.gov/dms/ (last access: 1 April 2020) | In situ | October 1980–August 2017 | – |
| | Copernicus-GlobColour, Level 4 | https://data.marine.copernicus.eu/ product/OCEANCOLOUR_GLO_ BGC_L4_MY_009_104/description (last access: 25 February 2024) | Daily | September 1997–present | $0.042° \times 0.042°$ |
| | CMEMS global biogeochemical multi-year hindcast (only used for the simulation of DMS concentrations in polar regions when satellite Chl $a$ is unavailable) | https://data.marine.copernicus.eu/ product/GLOBAL_MULTIYEAR_ BGC_001_029/description (last access: 25 February 2024) | Daily | January 1993–present | $0.25° \times 0.25°$ |
| SST | NOAA OI SST V2 | https://psl.noaa.gov/data/gridded/data. noaa.oisst.v2.highres.html par (last access: 2 April 2020) | Daily | September 1981–present | $0.25° \times 0.25°$ |
| MLD | NASA ECCO V4r4 | https://data.nas.nasa.gov/ecco/data. php?dir=/eccodata/llc_90/ECCOv4/ Release4 (last access: 25 May 2020) | Daily | January 1992–December 2017 | LLC90 (22–110 km) |
| DSWF | | | | | |
| SSS | | | | | |
| Nitrate | CMEMS global biogeochemical multi-year hindcast | https://data.marine.copernicus.eu/ product/GLOBAL_MULTIYEAR_ BGC_001_029/description (last access: 25 February 2024) | Daily | January 1993–present | $0.25° \times 0.25°$ |
| Phosphate | | | | | |
| Silicate | | | | | |
| DO | | | | | |
| WS | NASA ECCO V4r4 | https://data.nas.nasa.gov/ecco/data. php?dir=/eccodata/llc_90/ECCOv4/ Release4 (last access: 25 May 2020) | Daily | January 1992–December 2017 | LLC90 (22–110 km) |
| SI | NOAA OI SST V2 | https://psl.noaa.gov/data/gridded/data. noaa.oisst.v2.highres.html (last access: 2 April 2020) | Daily | September 1981–present | $0.25° \times 0.25°$ |

tificial neural network (ANN) model. The target feature was $\log_{10}(\text{DMS})$, and the input features were $\log_{10}(\text{Chl }a)$, SST, $\log_{10}(\text{MLD})$, $\log_{10}(\text{nitrate})$, $\log_{10}(\text{phosphate})$, $\log_{10}(\text{silicate})$, DO, DSWF, and SSS. The data for all variables were standardized before training.

We randomly selected 10 % of the samples ($n = 4116$) to be entirely excluded from training as a testing subset for global validation and the overfitting test. Specifically, 401 samples were randomly selected from the Coastal biome and 3715 samples were selected from other biomes to compose the testing subset, which matched the proportion of the global ocean that is coastal (9.7 %). Then, the remaining samples ($n = 37041$) were utilized for training and cross-validation, with the constraint that $F_{\text{coastal}}$ was equal to 9.7 % in each training session, as mentioned above.

Our feedforward fully connected neural network comprised two hidden layers, with 15 nodes in each layer. The activation functions for the first and second layers were ReLU and tanh, respectively. We applied L2 regularization (lambda $= 1 \times 10^{-4}$ TS1) to counteract overfitting. The loss function was the mean square error (MSE). Training stopped if the validation loss was greater than or equal to the minimum validation loss computed so far 20 times in a row. The training processes were carried out with the Statistics and Machine Learning Toolbox in MATLAB 2022b. We repeated the data split (for training and validation sets) and training processes 100 times and obtained 100 neural networks. The average prediction results from multiple ANNs show a much higher consistency with the observations than obtained with a single ANN (Fig. S3). As the number of ANNs ($N_{\text{training}}$) increases, the accuracy of the model pre-

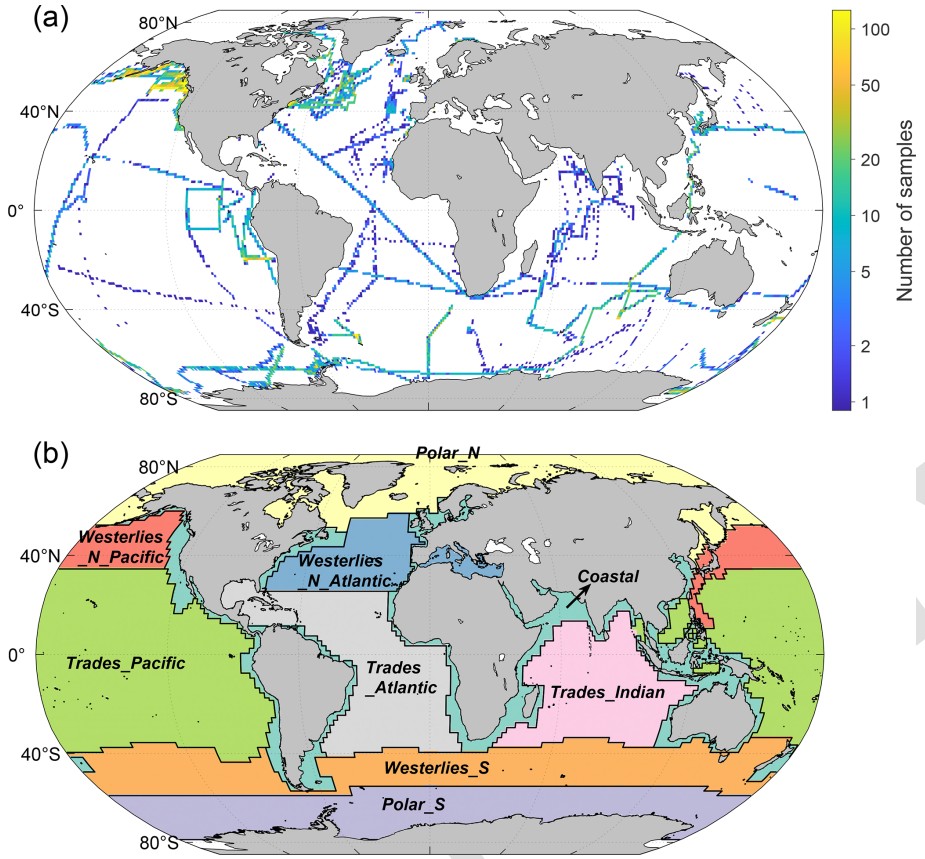

**Figure 2. (a)** The distribution of the 41 157 DMS observational data values after matchup, filtration, and binning when constructing the ANN model. The grid size is $1° \times 1°$. **(b)** The nine oceanic regions that were separated based on Longhurst's biomes (Longhurst, 1998).

dictions initially improves and then stabilizes. We adopted the average output of 20 ANNs as the final output, balancing performance and computational costs effectively. This kind of multiple-training approach, often termed an "ANN ensemble" or "Monte Carlo cross-validation", has been widely used to improve model generalization and performance (Sigmund et al., 2020; Holder et al., 2022) as well as to get a better model evaluation (Dubitzky et al., 2007).

## 2.4 Deriving the 20-year global DMS distributions

### 2.4.1 Simulation of sea surface DMS concentrations

First, we constructed the daily gridded dataset of input variables with a spatial resolution of $1° \times 1°$ from 1998 to 2017 based on the data sources listed in Table 1 (except the in situ Chl $a$ data). Datasets with a higher spatial resolution than $1° \times 1°$ were binned into $1° \times 1°$. Satellite Chl $a$ data for the polar regions obtained during winter were missing, so the Chl $a$ data from the CMEMS global biogeochemical multiyear hindcast were used to fill in the missing values. Then, the obtained gridded dataset was fed into the ANN ensemble model, and the 20-year global distribution of sea surface DMS concentration with daily resolution was simulated.

### 2.4.2 Calculation of sea-to-air fluxes

The sea-to-air fluxes of DMS were calculated on the basis of simulated surface DMS concentrations using Eq. (1):

$$\text{DMS flux} = \text{Kt} \times \left( \text{DMS}_w - \frac{\text{DMS}_a}{H} \right). \tag{1}$$

Here, $\text{DMS}_w$ and $\text{DMS}_a$ are the DMS concentrations in surface seawater and air, respectively. $H$ is the Henry's law constant for DMS. Since $\frac{\text{DMS}_a}{H}$ is usually $\ll \text{DMS}_w$, this term was omitted in the calculation. Kt is the total transfer velocity considering the sea ice coverage fraction (SI):

$$\text{Kt} = k_t \times (1 - \text{SI}). \tag{2}$$

$k_t$ is the total transfer velocity without considering sea ice, which is calculated by Eq. (3):

$$\frac{1}{k_t} = \frac{1}{k_w} + \frac{1}{k_a \times H}. \tag{3}$$

Here, $k_w$ and $k_a$ are the water-side transfer velocity and air-side transfer velocity, respectively. We used the same approach as Galí et al. (2019) to obtain $k_w$, $k_a$, and $H$ for DMS, where the effect of wind speed was considered for

$k_a$ and the influences of SST and SSS were considered for $H$. The calculations of $k_a$ and $H$ followed the parameterizations of Johnson (2010). To calculate $k_w$, we adopted the bubble scheme (Woolf, 1997), which divided the sea-to-air mass transfer process into turbulence- and bubble-mediated gas exchange. The $k_w$ calculated based on the bubble scheme is lower than that from Nightingale's scheme (Nightingale et al., 2000) under the conditions of a high wind speed, and it exhibits a smaller deviation from the measurements (Beale et al., 2014; Galí et al., 2019). Before the calculation, the WS and SI data were also binned into a $1° \times 1°$ grid. By using WS and SI together with SST and SSS datasets, we obtained the daily gridded Kt and then calculated the sea-to-air DMS fluxes (daily; 1998–2017) by multiplying the simulated DMS concentrations by the Kt values.

## 3 Results

### 3.1 Model performance

As shown in Fig. 3a, the newly developed ANN ensemble model captures a substantial part of the data variance globally ($\log_{10}$ space $R^2 = 0.651$ and RMSE $= 0.262$). A total of 92.8 % of the ANN-simulated concentration values fall within 1/3 to 3 times the corresponding true value. The performance for the testing set ($R^2 = 0.640$, RMSE $= 0.267$, and 92.7 % of data within the range of 1/3 to 3 times the observed value) is very close to that for the training set (Fig. 3b), suggesting no obvious overfitting. The ANN model exhibits better performance compared to previous empirical and process-based models ($R^2 = 0.01$–$0.14$) (Tesdal et al., 2016) as well as the satellite-based algorithm ($R^2 = 0.50$) (Galí et al., 2018). Compared to our model, the ANN model developed by Wang et al. (2020) showed a similar performance ($R^2 = 0.66$ and RMSE $= 0.264$ for the training set) despite its more complex ANN configuration (two hidden layers with 128 nodes each) and the inclusion of sample location and time among its input features. However, the greater complexity of that model will significantly increase its computational cost, and the incorporation of location and time information may weaken the physical interpretability.

The performance of the model was evaluated across each of the nine oceanic regions. As illustrated in Figs. 3c and 4, the $\log_{10}$ space RMSEs are all below 0.32 (equivalent to a concentration ratio of 2.09 in linear space), except for the Coastal region (training: RMSE $= 0.322$, $R^2 = 0.479$; testing: RMSE $= 0.332$, $R^2 = 0.480$). Since the Coastal region comprises only 9.7 % of the global oceanic area, the comparatively low performance in this area has a minimal impact on the overall ability to predict the spatiotemporal distribution of DMS on a global scale. Despite the $R^2$ values in Trades_Pacific and Trades_Atlantic being lower than 0.5, which is related to the relatively narrow range of DMS concentration variation, the RMSEs in these regions remain quite low and comparable to those in other regions. In general, our

ANN ensemble model demonstrates a satisfactory capacity to reproduce variations in DMS concentrations across diverse oceanic regions.

However, it is noteworthy that our model tends to underestimate extremely high DMS concentrations and overestimate extremely low concentrations. Overall, the linear regressions between ANN-predicted and observed DMS concentrations yield slopes that are significantly lower than unity across all regions (Figs. 3c and 4), and there are significantly positive correlations between prediction residuals (observation − prediction) and the observed $\log_{10}$(DMS) (Figs. S5 and S6). From a data perspective, this may be partly due to the insufficient number of samples with extreme DMS concentrations (known as underrepresentation), making it difficult to adequately capture the relevant information during the training process. To test this point, we adopted a weighted resampling strategy to bolster the number of samples in the minority class before training. This strategy has been widely used in machine learning to deal with the data imbalance issue (Haibo et al., 2008; Yu and Zhou, 2021; Chawla et al., 2002). The basic idea is to set a higher probability of being sampled for the minority class with extreme DMS concentrations; the details are illustrated in Fig. S7 and explained in Appendix B. The results indicate that the weighted resampling scheme cannot fully alleviate the model bias. Although it does elevate the overall prediction-versus-observation slope from $\sim 0.59$ to $\sim 0.63$, this improvement is marginal (Figs. S8 and S9). In several regions like the Westerlies_S and Trades biomes, the slopes are even lower than the original values. Furthermore, the data become more scattered after implementing the weighted resampling, resulting in an increased RMSE and decreased $R^2$. Therefore, there are other potential issues causing the model bias, which are discussed in Sect. 4. The original model, trained without weighted resampling, was adopted for subsequent analysis and the construction of the gridded DMS dataset.

Primarily owing to the underestimation of high DMS concentrations, a negative mean bias (MB) in DMS concentration is evident across all regions, ranging from $-0.18$ to $-2.02$ nM (Table 2). The normalized mean bias (NMB; the ratio between mean bias and mean observed concentration) ranges from $-8.7$ % to $-32.2$ %. The most significant NMB emerges in Coastal and Trades_India regions, while the NMB remains within $-25$ % for other regions. The global MB and NMB are $-1.05$ nM and $-22.1$ %, respectively. It is worth noting that these biases are compared against historical DMS observations, which were conducted within a very limited geographical area and very limited time periods. Thus, they cannot be interpreted as the actual mean modeling bias for the entire region. On the other hand, the negative biases at the high end of the concentrations are partially canceled out by the positive biases at the low end during the averaging over the entire region. The bias at a specific grid could be much larger. Nevertheless, those extreme DMS concentrations ($>15$ nM or $<0.3$ nM) that exhibit the most significant

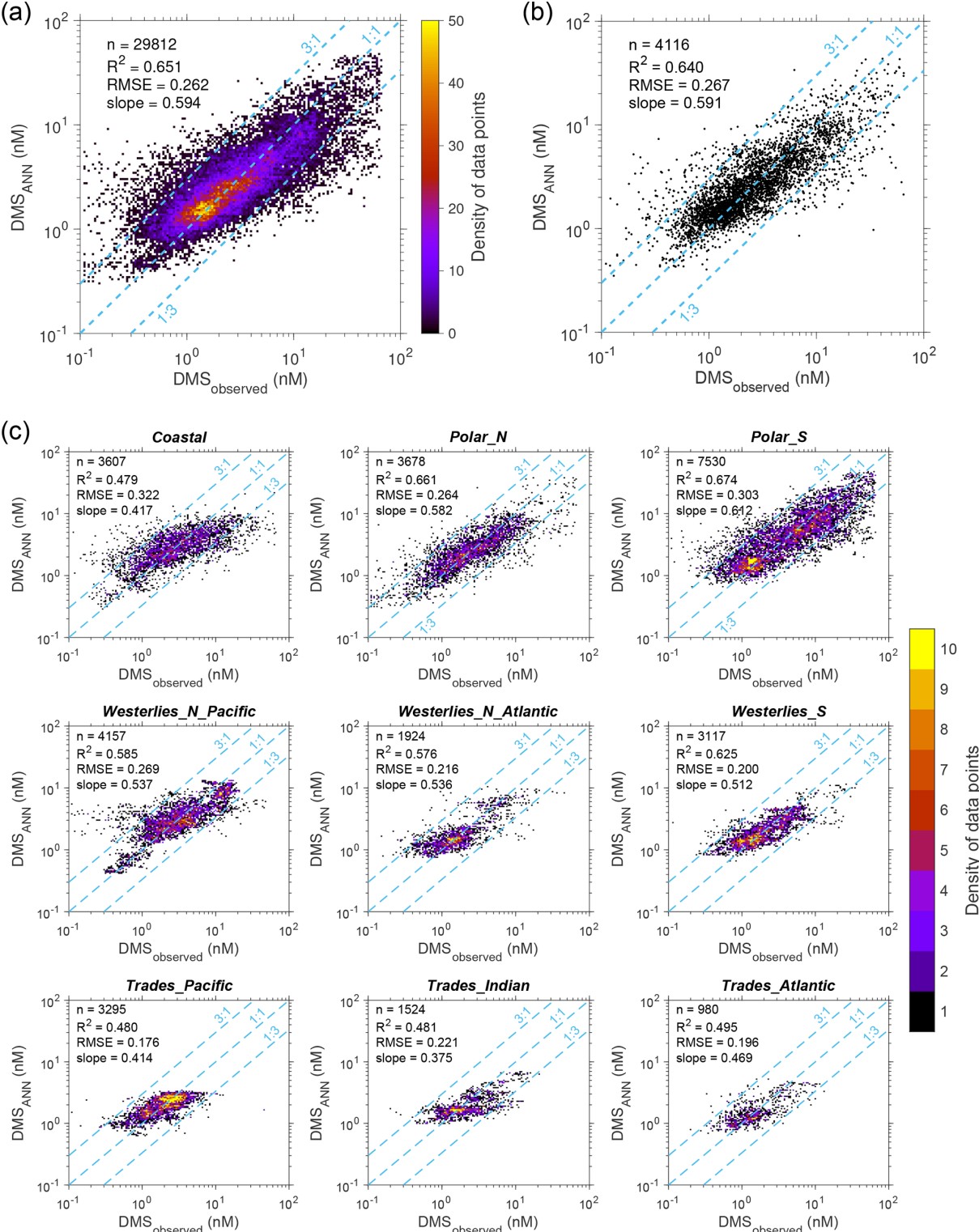

**Figure 3.** Comparisons between ANN-simulated and observed DMS concentrations. **(a)** Scatter density for simulated versus observed DMS concentrations of the samples used in ANN training. **(b)** Comparison between the simulated and observed DMS concentrations in the testing set. **(c)** Comparison between the simulated and observed DMS concentrations for the samples used in ANN training across nine regions. The number of data points ($n$), $\log_{10}$ space $R^2$, root mean square error (RMSE), and linear regression slope are also displayed.

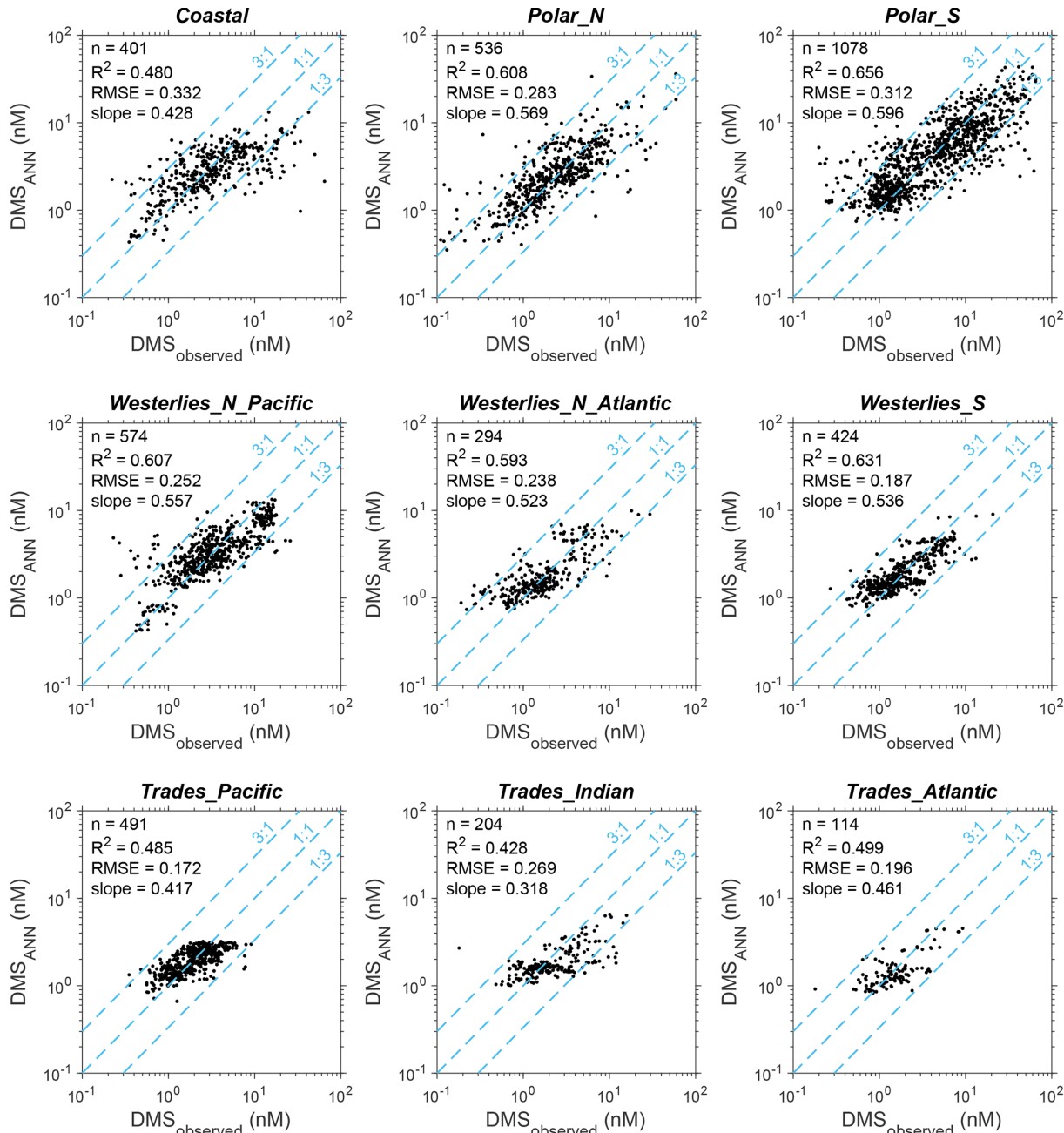

**Figure 4.** Comparisons between the simulated and observed DMS concentrations in the testing set across nine regions.

modeling bias represent only a minority of the entire sample set (6.9 %). Our model adeptly reproduces the majority of observations with moderate DMS concentrations across all regions, with the percentage of predicted values falling within 1/3 to 3 times the observed value ranging from 87.0 % to 98.8 %.

It is worth noting that there may be intrinsic connections between the 10 % of samples excluded as a testing subset and the training set because the data from the same cruise or fixed-site campaign have a certain level of continuity. To further evaluate the reliability of the ANN model, we compared the simulated DMS concentrations with the observational data from fully independent campaigns. The latter data were obtained from 33 cruises in the northeastern Pacific, western Pacific, and North Atlantic (number of data values 6478). These data include (1) discrete samplings and measurements during 31 cruises of the Line P program in the northeastern Pacific (Steiner et al., 2011) (9 February 2007– 26 August 2017; number of data values 177; https://www. waterproperties.ca/linep/index.php, last access: 23 Novem-

**Table 2.** The mean bias and normalized mean bias of the ANN-predicted DMS concentrations against observations across different regions.

| Region | Mean bias (nM) | Normalized mean bias |
|---|---|---|
| Coastal | −1.55 | −32.2 % |
| Polar_N | −0.90 | −21.4 % |
| Polar_S | −2.02 | −24.1 % |
| Westerlies_N_Pacific | −0.91 | −18.8 % |
| Westerlies_N_Atlantic | −0.24 | −10.4 % |
| Westerlies_S | −0.36 | −14.1 % |
| Trades_Pacific | −0.19 | −8.7 % |
| Trades_Indian | −0.73 | −26.7 % |
| Trades_Atlantic | −0.18 | −10.1 % |
| Global | −1.05 | −22.1 % |

ber 2020), (2) underway measurements performed during SONNE cruise 202/2 (TRANSBROM) in the western Pacific (Zindler et al., 2013a) (9–23 October 2009; number of data values 115; https://doi.org/10.1594/PANGAEA.805613, Zindler et al., 2013b), and (3) underway measurements performed during the third North Atlantic Aerosols and Marine Ecosystems Study (NAAMES) campaign (Behrenfeld et al., 2019; Bell et al., 2021) (6–24 September 2017; number of data values 1025; https://seabass.gsfc.nasa.gov/naames, last access: 27 November 2020). Before the comparison, the data measured within a $0.05° \times 0.05°$ grid on the same day were binned by arithmetic averaging.

The comparisons between these observed DMS concentrations and the ANN simulation are shown in Fig. 5. Regarding the Line P program, it should be noted that seven cruises are included in the GSSD database, but those data were obtained by underway measurements, different from the discrete sampling (Niskin bottle) data used here. Hence, these cruises were retained and are marked in Fig. 5a but excluded in the subsequent statistical analysis (Fig. 5b, c). It can be seen that the model effectively captures the seasonal variation in the northeastern Pacific, which is generally August > June > February (Fig. 5a). However, the small-scale spatial variations are only partially reproduced by the model in certain campaigns, such as those performed in June and August of 2007, June of 2009, August of 2012, and August of 2016. Notably, the model generally underestimates high DMS concentrations during summer, particularly those exceeding 10 nM, consistent with earlier discussions. Aggregating data from all campaigns across three regions, the $\log_{10}$ space RMSE of the simulated DMS concentrations against the observations is 0.274, marginally higher than for the training set. Most simulated values (93.0 %) are within the range of 1/3 to 3 times the observed value. The results further evidence that there is no significant overfitting in our model. When data from each campaign are binned, simulations demonstrate high consistency with observations, as

depicted in Fig. 5c (RMSE = 0.249; $R^2 = 0.758$). In summary, although our ANN ensemble model may not precisely reproduce small-scale variations and extreme values in specific regions and periods, it captures the overall large-scale variations reasonably well.

## 3.2 DMS distribution

### 3.2.1 Spatial and seasonal variations

The monthly climatology of ANN-simulated DMS concentrations in the global sea surface from 1998 to 2017 is shown in Fig. 6. Overall, the DMS concentrations in mid- and high-latitude regions exhibit a significant seasonal cycle, peaking in summer and reaching their lowest in winter. This pattern aligns with the results of many prior observational studies. In the Northern Hemisphere, elevated DMS concentrations (>2.5 nM) during summer mainly occur in two regions. One is the North Pacific (40–60° N), where the concentration generally peaks in August, surpassing 10 nM (Fig. 6). The other is the subarctic North Atlantic (45–80° N). A notable increase in DMS concentration starts at around 45–50° N in May and gradually shifts northward beyond 50° N by July (Figs. 6–7). This spatiotemporal evolution pattern corresponds to the evolution of solar radiation intensity and the spring–summer bloom patterns of phytoplankton (Friedland et al., 2018; Yang et al., 2020). The peak concentration date at the same latitude in the North Atlantic generally precedes that in the North Pacific (Fig. 7). In the Southern Hemisphere, there is a conspicuous DMS-rich zone near 40° S (where the Subtropical Convergence lies) in summer. This presents as a ring-shaped high-concentration band that is nearly parallel to that latitude. The highest seasonal mean concentration (December–February) occurs at 41.5° S, reaching 3.71 nM (Fig. 9). Southward from this zone, there is a low-DMS area spanning 47–61° S where the average concentration is below 2.5 nM across all seasons. However, in the coastal waters of Antarctica (south of 60° S), significantly high concentrations also manifest in summer; these surpass 4.0 nM, even higher than those near 40° S (Figs. 6 and 9). In addition to the above regions, several typical upwelling zones also exhibit relatively high DMS concentrations, such as the eastern Pacific and the southeastern Atlantic. The former, situated at lower latitudes, shows no significant seasonal variation, while the latter exhibits higher concentrations from October to February. The high nutrient concentrations in upwelling areas can bolster primary productivity, intensifying biological activities and augmenting the production of biogenic sulfur.

The spatiotemporal variation of DMS emission flux is generally consistent with that of the concentration. As shown in Fig. 8, DMS fluxes are also significantly higher in summer across most mid- and high-latitude regions, and the high-flux regions generally overlap with the hot spots of DMS concentration. This indicates that the distribution of sea sur-

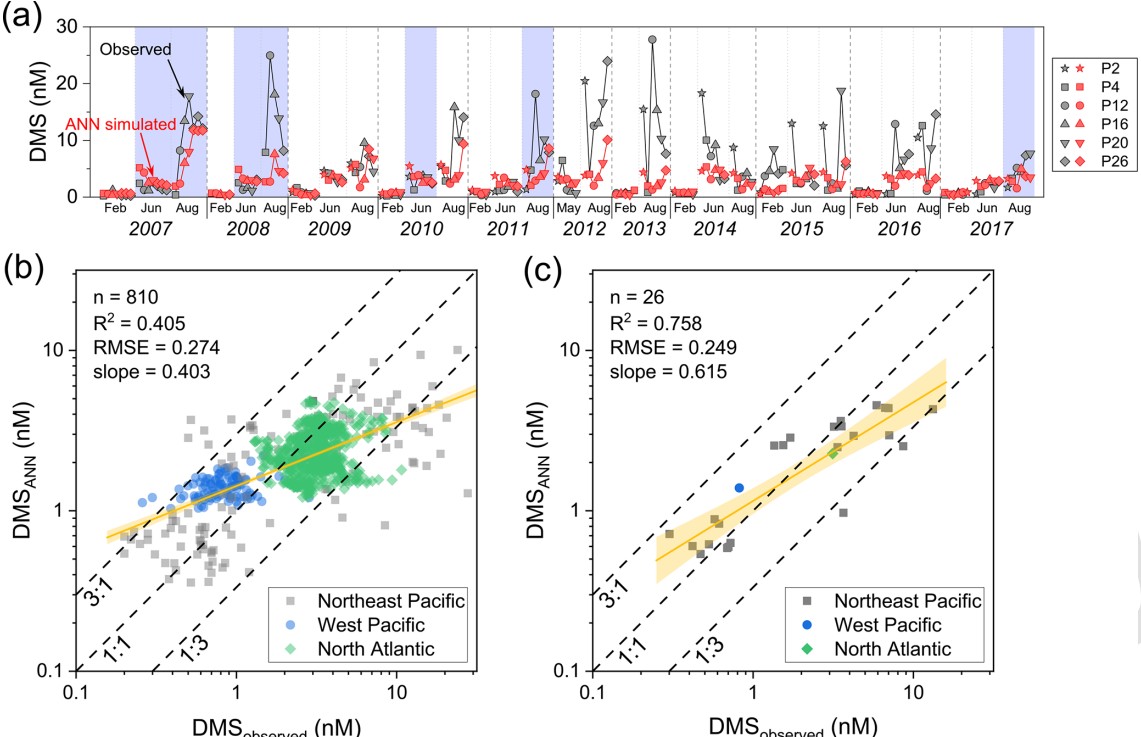

**Figure 5.** Comparisons between the ANN predictions and observations from fully independent campaigns. **(a)** Time series of simulation results and DMS observational data obtained from the Line P program. The different markers represent different stations along Line P. The blue shading covers the cruises included in the GSSD database. **(b)** Scatter plot of simulated versus observed DMS concentrations. **(c)** The same as panel **(b)** but for the averaged data from each cruise. The yellow lines and shaded bands are linear fits and the corresponding 95 % confidence intervals for $\log_{10}$ space data. The values of $R^2$, RMSE, and slope displayed in the figure also correspond to the $\log_{10}$ space data.

face DMS concentration is the main factor controlling the monthly variation pattern of DMS emissions at the global scale, and the effect of transfer velocity is secondary. However, certain regions present inconsistencies between DMS flux and the concentration dynamics. For instance, in the Arabian Sea and the central Indian Ocean, elevated transfer velocities (Fig. S10) during June to September, driven by heightened wind speeds, markedly enhance emission fluxes despite the comparatively low concentrations compared to other months. In polar regions, especially along the coast of Antarctica, although the DMS concentration is high in summer, sea ice coverage significantly impedes DMS release; thus, the emission flux remains at a low level.

As shown in Fig. 9, the higher wind speeds in autumn and winter at mid-latitudes and high latitudes result in higher total transfer velocities, leading to smaller summer-to-winter ratios of DMS emission flux compared to that of DMS concentration. At low latitudes, the existence of the trade wind zones in both hemispheres further leads to two high-flux bands. The emission fluxes in the equatorial region between these two trade zones are significantly lower. Although the latitudinal distributions of mean DMS emission flux in the Southern and Northern Hemisphere are almost symmetrical, the huge difference in ocean area between the two hemi-

spheres results in significantly higher total emissions from the Southern Hemisphere. Since anthropogenic $SO_2$ emissions are mainly concentrated in the Northern Hemisphere, oceanic DMS plays a much more important role in the Southern Hemisphere, especially over the regions south of $40°$ S where the DMS emissions are high and the perturbation of anthropogenic pollution is low.

According to our newly built DMS gridded dataset, the global area-weighted annual mean concentration of DMS at the sea surface from 1998 to 2017 was $\sim 1.71$ nM (1.67–1.75 nM), which is within the range of values (1.6 to 2.4 nM) obtained by various methods in previous studies (Tesdal et al., 2016). The global annual mean DMS emissions into the atmosphere were 17.2 Tg S yr$^{-1}$ (16.9–17.5 Tg S yr$^{-1}$), with 10.3 Tg S yr$^{-1}$ (59.9 %) originating from the Southern Hemisphere and 6.9 Tg S yr$^{-1}$ (40.1 %) from the Northern Hemisphere.

### 3.2.2 Comparisons with other global DMS climatologies

Here we compare the distributions of DMS concentration derived from our ANN simulation (referred to as Z23) with four previously constructed climatologies (Fig. 10), including (1) L11 (the widely used second version of the interpolation-

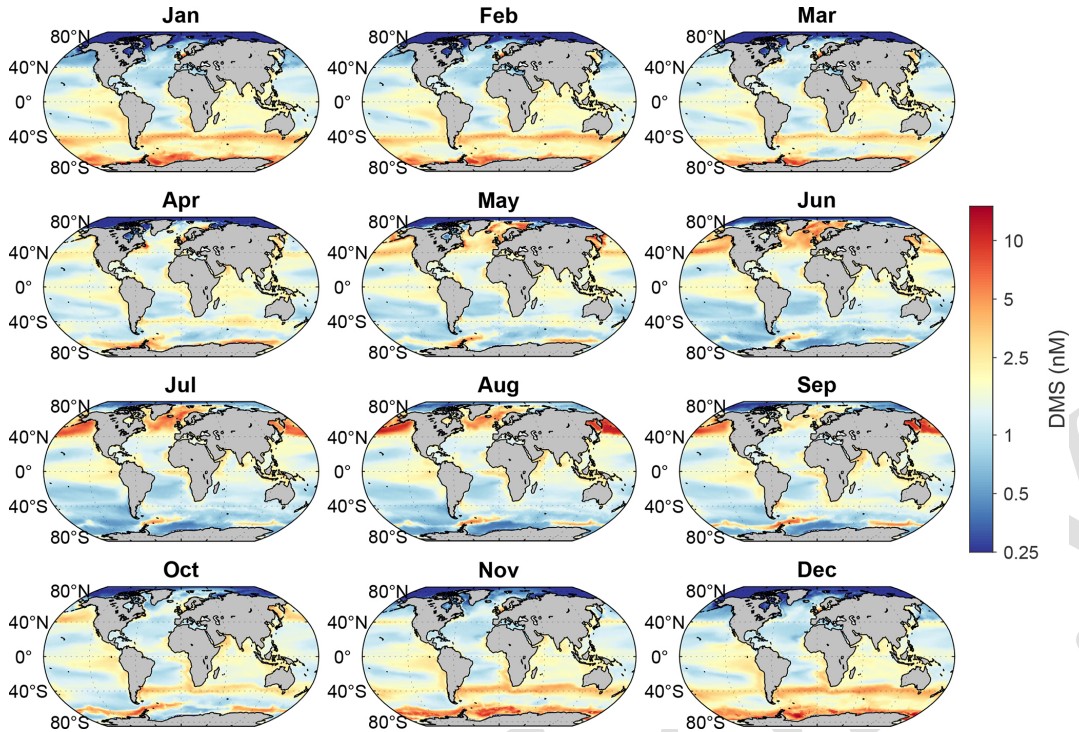

**Figure 6.** Monthly climatology of global sea surface DMS concentration during 1998 to 2017.

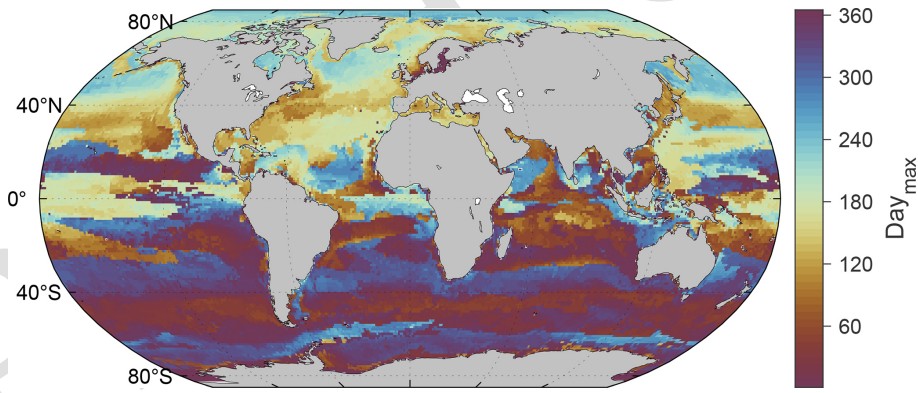

**Figure 7.** The day of the year with the highest sea surface DMS concentration for each grid point.

and/or extrapolation-based climatology established by Lana et al., 2011), (2) H22 (an updated version of L11 that incorporates many more DMS measurements and uses dynamic biogeochemical provinces; Hulswar et al., 2022), (3) G18 (the DMS concentration field estimated by a two-step remote sensing algorithm; Galí et al., 2018), and (4) W20 (the previous DMS climatology simulated by an ANN; Wang et al., 2020).

Overall, all datasets exhibit the general pattern of high DMS concentrations during summer and low concentrations during winter, but notable distinctions between their specific distributions emerge. Due to the limitation of the method used, $DMS_{L11}$ exhibits relatively low spatial heterogeneity (i.e., higher patchiness), which may not capture the detailed spatial variability on a regional scale well. Compared with $DMS_{L11}$, $DMS_{Z23}$ is significantly lower at high latitudes during summer and in the southern Indian Ocean and southwestern Pacific Ocean from December to February (Fig. 10a). Particularly in the southern polar region (Polar_S), latitudinal averages of $DMS_{L11}$ surpass 10 nM during summer, which is 1–3 times higher than those of $DMS_{Z23}$ (Fig. 10e). However, $DMS_{Z23}$ maintains a similar level around the Antarctic in March compared to summer, and it is significantly higher than $DMS_{L11}$ as well as the other three climatologies. $DMS_{H22}$ shows lower disparities with $DMS_{Z23}$ in the Arctic, the southern Indian Ocean, and the southwestern Pacific

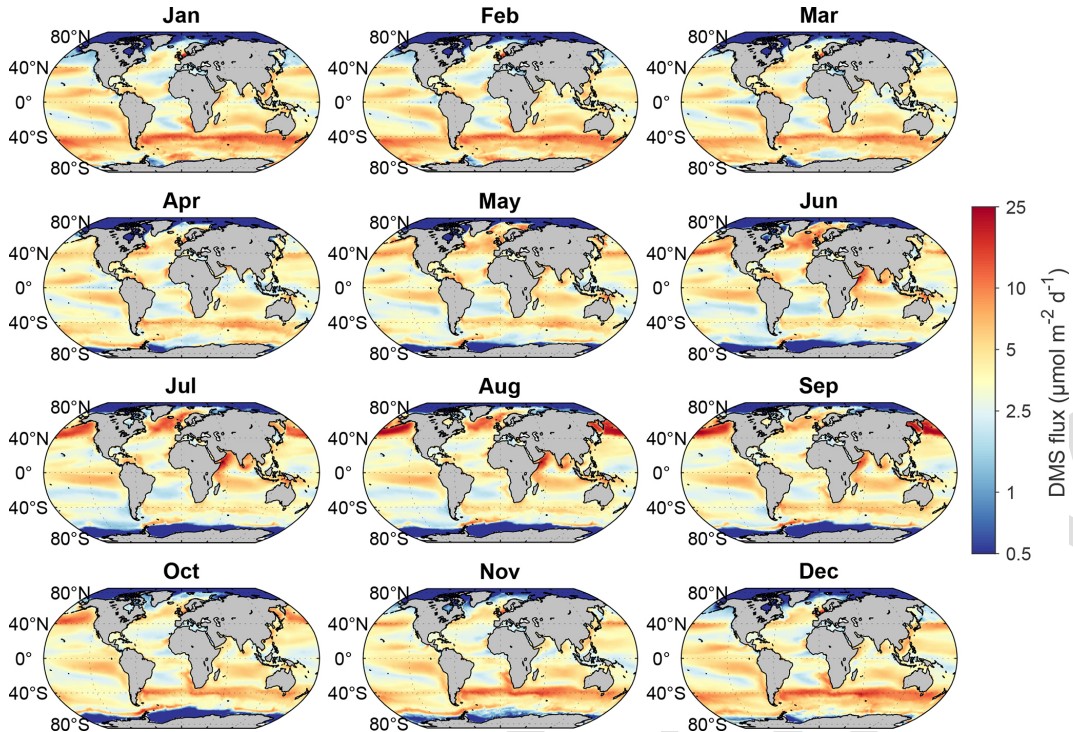

**Figure 8.** Monthly climatology of global DMS sea-to-air flux from 1998 to 2017.

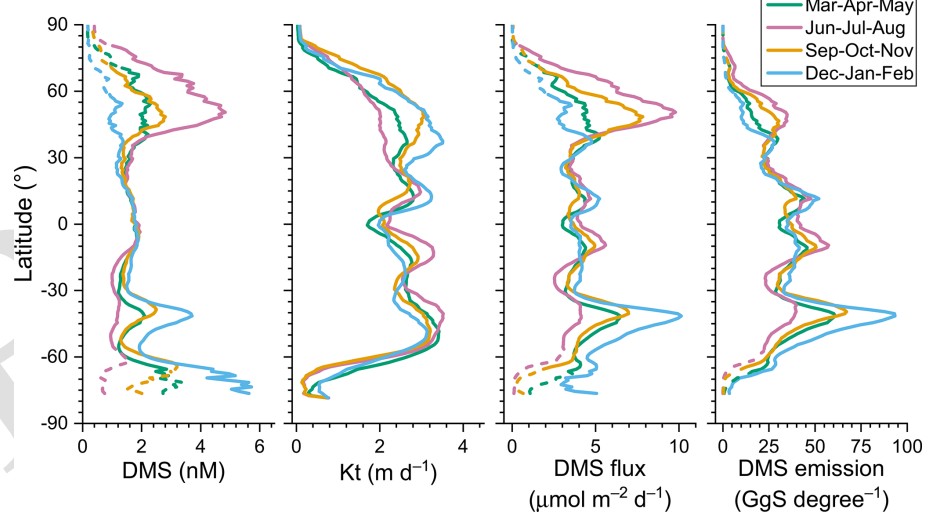

**Figure 9.** Latitudinal distributions of sea surface DMS concentration, total transfer velocity (Kt), sea-to-air flux, and total emissions in different seasons during 1998–2017. The dashed parts of the lines indicate regions where more than half of the satellite Chl $a$ values were missing and thus not available for the DMS simulation, so most of the Chl $a$ data for these regions are from the CMEMS global biogeochemical multi-year hindcast.

Ocean, but the summertime concentrations in most of the Polar_S region are also >2 nM higher than $DMS_{Z23}$ (Fig. 10b). In contrast, $DMS_{H22}$ in Polar_S from September to November is ∼ 2 nM lower than $DMS_{Z23}$. The global area-weighted annual mean DMS concentrations in L11 and H22 are 2.43 and 2.26 nM, respectively, which are approximately 42.1 % and 32.2 % higher than Z23.

G18 exhibits the lowest global annual mean concentration (1.63 nM) among these climatologies, approximately 4.7 % lower than Z23. The most notable deviation occurs in the North Pacific during boreal summer and near the Antarctic

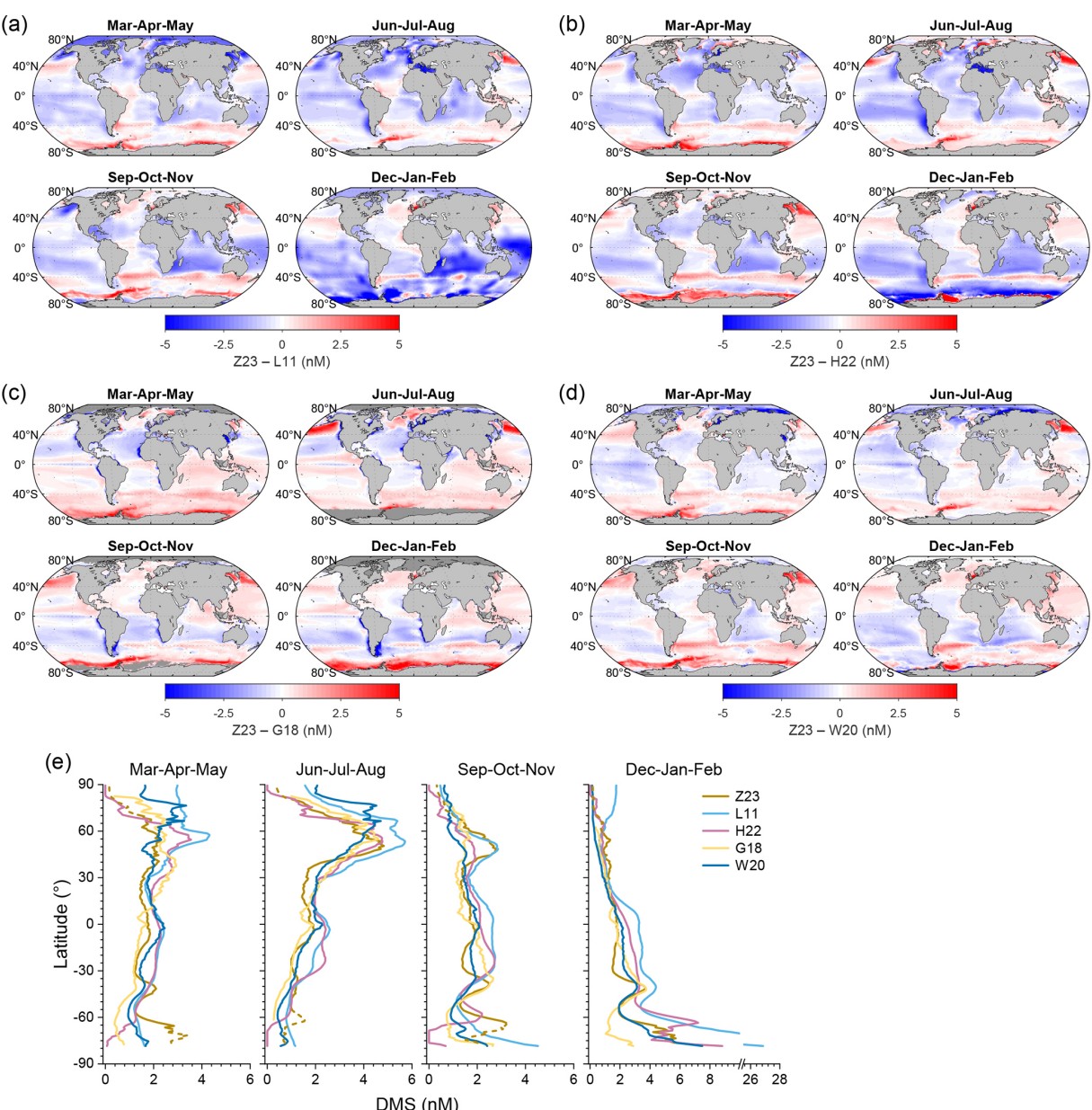

**Figure 10. (a–d)** The spatial distributions of DMS concentration differences between Z23 and four previously estimated fields across different seasons: **(a)** L11, **(b)** H22, **(c)** G18, and **(d)** W20. Dark gray regions of the ocean represent areas where data are missing for at least one field. **(e)** Comparisons between the latitudinal distributions of Z23 and four previous DMS fields across different seasons. The dashed parts of the Z23 lines indicate regions where more than half of the satellite Chl $a$ values were missing and thus not available for the DMS simulation, so most of the Chl $a$ data for these regions are from the CMEMS global biogeochemical multi-year hindcast.

during austral summer; in these cases, $DMS_{Z23}$ is >3.5 nM (>100 %) higher than $DMS_{G18}$ (Fig. 10c). Conversely, there are high DMS concentrations (>5 nM) in certain coastal seas (such as the coasts of eastern and northeastern Asia, the coasts of Patagonia and Peru, the southwestern coast of Africa, and the western coasts of the Sahara and North America) based on the G18 estimate. This characteristic is not fully replicated by other DMS fields, possibly due to the underestimation of DMS by our model and other meth-

ods in coastal regions as well as the overestimation of Chl $a$ by satellites, which is caused by interference from colored dissolved organic matter and non-algal detrital particles (Aurin and Dierssen, 2012). W20 exhibits the highest consistency with Z23 in spatiotemporal distribution patterns as well as the lowest difference in global annual mean concentration (1.74 nM, only 1.8 % higher than Z23). However, notable discrepancies exist in specific regions. For instance, during summertime, $DMS_{Z23}$ is >1 nM (>40 %) lower than

$DMS_{W20}$ in more than half of the Arctic area, while in the North Pacific and Southern Ocean, $DMS_{Z23}$ is significantly higher than $DMS_{W20}$ (Fig. 10d). Furthermore, only $DMS_{Z23}$ forms a nearly complete high-concentration annular band at $\sim 40°$ S during austral summer.

### 3.2.3 Decadal changes

One of the advantages of our ANN-derived DMS dataset is its time-resolved nature, which enables us to investigate the interannual variations in sea surface DMS concentration and flux. Here we present the decadal trends in DMS concentration, Kt, and emission flux spanning from 1998 to 2017 at both global and regional scales. Overall, the absolute inter-annual variability of DMS concentration across most global oceanic regions appears relatively small. A total of 88.4 % of the global oceanic area exhibited a range of less than 1 nM between the maximum and minimum annual average concentrations during this 20-year period. This was particularly evident in tropical and subtropical regions with latitudes between 40° S and 40° N. At latitudes higher than 40° in both hemispheres, notable decadal changes occurred (Fig. 11a). Annual mean DMS concentrations in the Greenland Sea, the North Pacific, and the Southern Ocean exhibited significant decreasing trends, with rates exceeding $0.03\,\mathrm{nM\,yr^{-1}}$ ($P<0.05$). A significant decreasing trend was also noted in the eastern tropical Pacific Ocean, albeit at a much lower absolute rate, primarily below $0.015\,\mathrm{nM\,yr^{-1}}$. Conversely, there were significant increasing trends in the Labrador Sea, the South Pacific (35–60° S, 150° E–75° W), and the southeastern Pacific, with the highest rate exceeding $0.02\,\mathrm{nM\,yr^{-1}}$. The global annual mean concentration exhibited a decreasing trend with a rate of $0.0035\,\mathrm{nM\,yr^{-1}}$ ($P<0.05$; Fig. 11d). The highest value (1.75 nM) occurred in 1999, and the lowest concentration (1.67 nM) occurred in 2015. Due to the primary influence of an increasing WS and the secondary impact of a rising SST in most mid- and low-latitude regions (Fig. S11), the Kt of DMS also showed an overall increasing trend, especially in the eastern Pacific and Atlantic Ocean (Fig. 11b). The increase in Kt can offset the decrease in DMS concentration to some extent, resulting in no significant trend in global DMS emissions during this 20-year period (Fig. 11d).

In the Arctic region, which is one of the most sensitive areas to climate warming (Screen et al., 2012; Serreze and Barry, 2011), the sea ice coverage has undergone a significant reduction over the past 2 decades; this is particularly noticeable in the Barents Sea and Kara Sea as well as further north ($>1\,\%\,\mathrm{yr^{-1}}$ for annual mean SI; Fig. S11). The retreat of summertime sea ice leads to an expansion of the open-sea surface, potentially amplifying DMS emission (Galí et al., 2019). However, despite this trend, there was no significant increase in the annual total emissions from the Polar_N region over the same period, primarily due to a decreasing trend in DMS concentration (Fig. 12). On the other hand, the highest emissions occurred in the last 2 years ($>0.64\,\mathrm{Tg\,S\,yr^{-1}}$) CE2, which are attributed to the highest Kt. Thus, with the further loss of sea ice coverage, it is likely that a rise in DMS emissions will appear in the Arctic region in the future (Galí et al., 2019). In contrast to the Arctic, the Southern Ocean has experienced a significant increase in the sea ice fraction (Fig. S11), leading to a significant decrease in Kt (Fig. 11b). Coupled with the decreased DMS concentration, this resulted in a substantial decline in the DMS emission flux (Figs. 11c and 12). The highest annual total emission flux in the Polar_S region occurred in 1998 (1.49 Tg S), while the lowest occurred in 2013 (1.02 Tg S), representing a decrease of $\sim 32\,\%$. Across other oceanic regions, the annual average DMS concentrations in the Westerlies_N_Pacific and Trades_Pacific regions exhibit decreasing trends over the past 20 years, while the concentration in Westerlies_S and Trades_Atlantic has increased ($P<0.05$; Fig. 12). Regarding DMS flux, Westerlies_N_Pacific showed a decrease, while Westerlies_N_Atlantic, Westerlies_S, and Trades_Atlantic showed an increase. There was no significant trend in other low-latitude regions.

### 3.3 Connection with atmospheric biogenic sulfur

One of the primary objectives of developing this daily gridded DMS dataset (Z23) spanning multiple years is to improve the emission inventory of marine biogenic DMS, thereby enhancing the modeling performance for atmospheric sulfur chemistry, especially when simulating sulfur-containing aerosols. To assess whether our newly constructed DMS dataset can reach this objective, we employed a backward-trajectory-based method to examine the correlation between sea surface DMS emissions and resulting DMS oxidation products in the atmosphere. The correlation was then compared against those derived from previously reported DMS climatologies (i.e., L11, H22, G18, and W20).

Here we use the observed concentrations of particulate methanesulfonic acid (MSA) over the Atlantic Ocean as a reference. MSA is one of the major end products of DMS in the atmosphere and derives solely from the oxidation of marine biogenic DMS over remote oceans (Saltzman et al., 1983; Savoie et al., 2002; Osman et al., 2019). Therefore, there is likely to be a dependence of the variation of MSA concentration on the DMS emission fluxes. During four transection cruises in the Atlantic conducted by the R/V *Polarstern* (20 April–20 May 2011, 28 October–1 December 2011, 10 April–15 May 2012, and 27 October–27 November 2012), the MSA concentrations in submicron aerosols were measured online using a high-resolution time-of-flight aerosol mass spectrometer. The ship tracks are shown in Fig. S12, and detailed information about the cruises and measurement methodology was provided by Huang et al. (2016). 72 h backward trajectories of air masses reaching the ship position were calculated every hour by the HYSPLIT model, starting from a height of 100 m (Stein et al.,

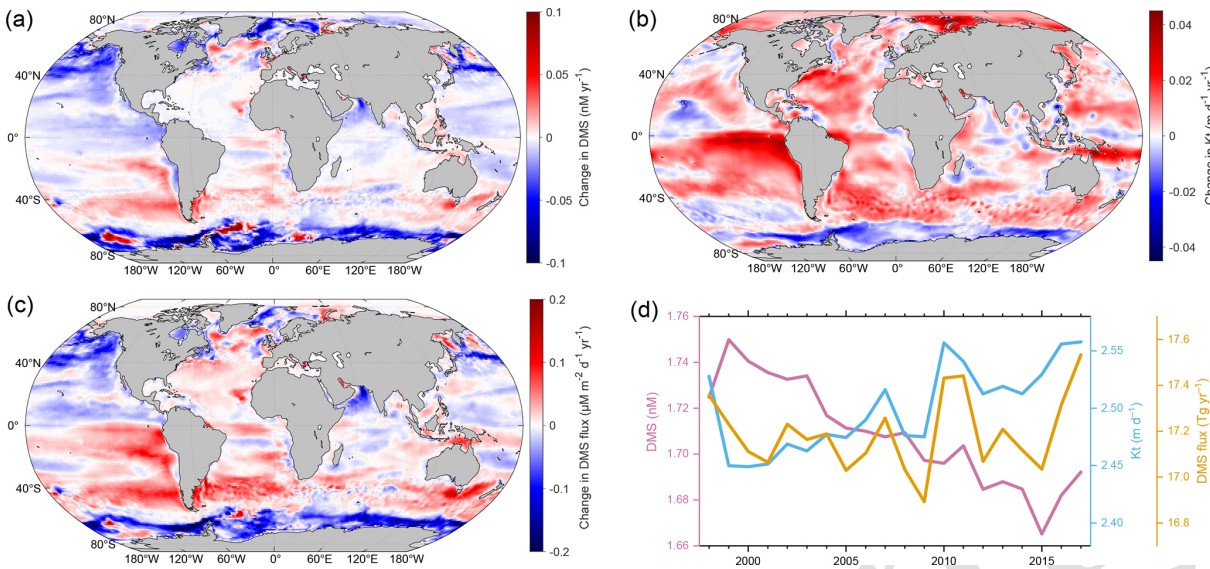

**Figure 11. (a–c)** The spatial distributions of changes in **(a)** DMS concentration, **(b)** Kt, and **(c)** DMS emission flux from 1998 to 2017. The linear regression slopes for the annual means are taken as the rates of change here. **(d)** The temporal changes in global annual mean DMS concentration, Kt, and total emission flux from 1998 to 2017.

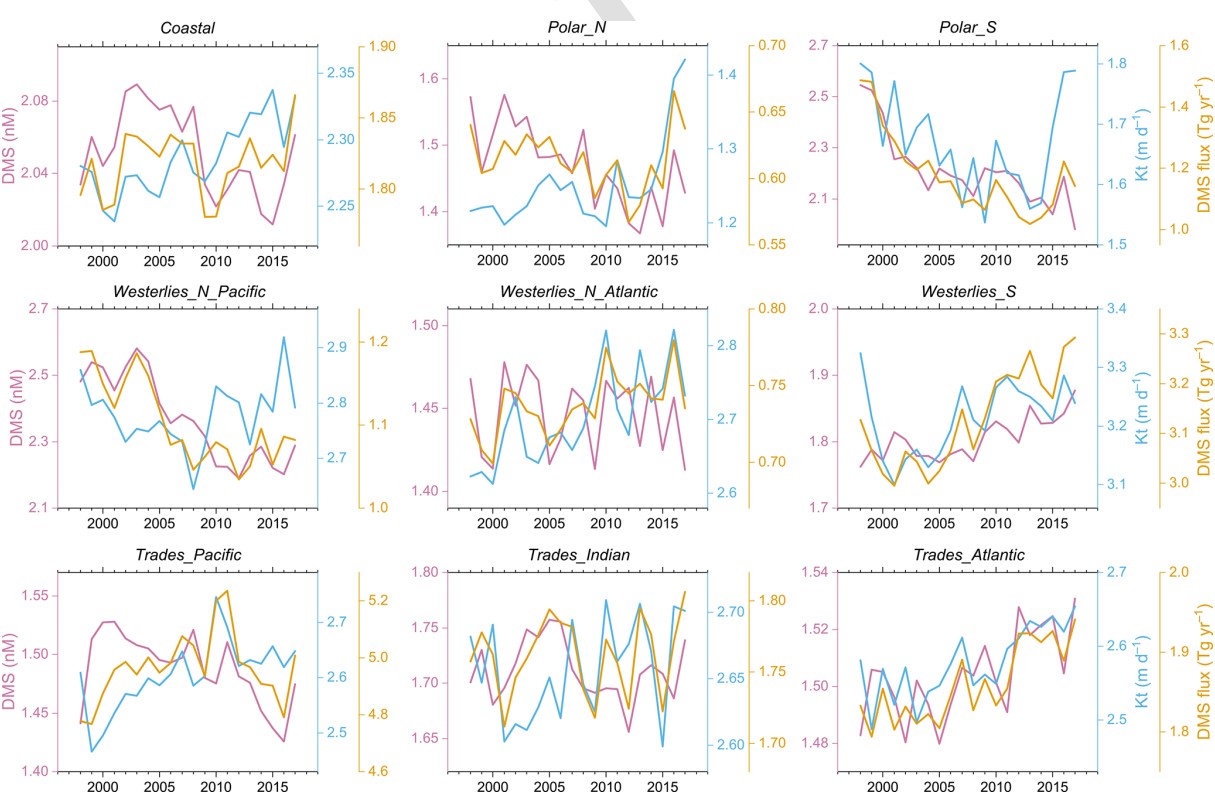

**Figure 12.** The temporal changes in annual mean DMS concentration, Kt, and total emission flux in different regions from 1998 to 2017.

2015). Subsequently, the air mass exposure to DMS emissions (AEDMS), denoting the weighted average of the DMS emission flux along the trajectory path, was calculated following the approach of Zhou et al. (2021). We used five different DMS gridded datasets, including Z23, L11, H22, G18, and W20. For Z23, the calculated daily DMS fluxes were utilized. For the remaining four monthly climatologies, we applied the daily Kt data from Z23 to calculate the DMS fluxes, thus eliminating the potential confounding influences stemming from different Kt parameterizations. In this calculation, the same concentration was assigned to all days within a month without interpolation. The detailed procedures for the calculation of AEDMS are elucidated in Appendix C.

MSA concentrations were significantly higher in late spring than in autumn for both the North and South Atlantic Ocean (Fig. 13a). For example, during the boreal spring cruise in 2011, the average MSA concentration over the North Atlantic ($0.068\,\mu g\,m^{-3}$ north of $25°N$) was about an order of magnitude higher than the average concentration over the South Atlantic ($0.006\,\mu g\,m^{-3}$ south of $5°S$). During the boreal autumn cruise in 2011, the average concentration over the South Atlantic ($0.034\,\mu g\,m^{-3}$ south of $5°S$) was $\sim 5$ times higher than that over the North Atlantic ($0.006\,\mu g\,m^{-3}$ north of $25°N$). In addition to this major seasonal pattern, there was also a minor MSA concentration peak between 5 and $15°N$ in both seasons. The spatial and seasonal variations in AEDMS based on the Z23 dataset (referred to as AEDMS_Z23) largely coincided with these MSA concentration patterns (Fig. 13a). It should be noted that the MSA / AEDMS ratio between 5 and $15°N$ was significantly lower than those in other high-MSA regions. This may result from the DMS simulation biases near the coast of West Africa or the lower DMS-to-MSA conversion yields, which are related to the air temperature and oxidant species (Barnes et al., 2006; Bates et al., 1992). There were also several AEDMS peaks in the North Atlantic during November 2012, which is inconsistent with the continuously low MSA concentrations. Given the high precipitation rates along the trajectory (Fig. 13a), a strong wet scavenging process might significantly reduce aerosol concentrations (Wood et al., 2017).

The AEDMS derived from other DMS concentration fields showed similar variations to AEDMS_Z23 (Fig. 13a). This is not surprising since all DMS concentration fields exhibit similar large-scale spatiotemporal patterns, and identical air mass transport paths and Kt values were applied in different AEDMS calculations. However, due to the lower temporal resolutions and the absence of interannual changes in those DMS monthly climatologies, the resulting AEDMS may be less effective in capturing variability at finer scales or across different years. To elaborate on this issue, here we focus on the high-MSA periods, which correspond to latitudes north of $25°N$ in boreal spring (S1 and S2 in Fig. 13a), $25°N$–$25°S$ in the boreal autumn of 2011 (A1 in Fig. 13a), and south of $5°N$ in the boreal autumn of 2012 (A2 in Fig. 13a). As shown in Fig. 13b, hourly MSA concentrations exhibited

significantly stronger correlations with AEDMS_Z23 than with other AEDMS time series in S1 and S2, indicating that AEDMS_Z23 can explain more (1.31–1.69 times more) of the variance of MSA concentration. During A1 and A2, the correlations between AEDMS and MSA concentration were weaker than those during S1 and S2, possibly due to higher DMS prediction biases in the South Atlantic or different influencing factors for atmospheric DMS chemistry across wide spatial ranges. Nonetheless, AEDMS_Z23 still exhibited the highest correlation with MSA (Fig. 13c). This overall stronger connection between Z23 and atmospheric DMS-derived aerosols mainly benefited from the combined effects of a higher time resolution and inherent interannual variations. For example, the ratio of the average MSA concentration during S1 to that during S2 (the S1-to-S2 ratio) was 1.89, and the A2-to-A1 ratio was 1.75. AEDMS_Z23 exhibited a slightly lower but still significant interannual variation, where the S1-to-S2 ratio and the A2-to-A1 ratio were 1.58 and 1.46, respectively. However, this interannual variation cannot be reproduced by other datasets, where the S1-to-S2 ratio and A2-to-A1 ratio were in the ranges of 1.08–1.30 and 1.19–1.29, respectively. These results show the potential of our newly developed DMS gridded data product to enhance the modeling performance for atmospheric DMS processes compared with previously reported climatologies.

It is worth noting that the satellite-based algorithms of G18 and the ANN model of W20 can also be utilized to produce daily multi-year DMS fields, just as Z23 does. Future investigations could include comparisons with these fields, facilitating a more comprehensive assessment of the performance of each algorithm or model. Furthermore, the AEDMS method used here is a highly simplified approach that does not consider the complex DMS chemistry in the atmosphere, and intercomparisons based on chemical transport models can be used in the future to obtain a more straightforward conclusion.

## 4   Uncertainties and limitations

Although our ANN ensemble model and derived DMS dataset demonstrate certain advantages compared to previous studies, as discussed in Sect. 3.3, notable uncertainties and limitations persist, resulting in $\sim 35\%$ uncaptured variance (Fig. 3a) and non-negligible simulation biases, e.g., the underestimation of extremely high DMS concentrations and the overestimation of low DMS concentrations. Firstly, there is a mismatch in the spatial and temporal scales between the input and target. The target, sea surface DMS concentrations, is obtained from in situ measurements taken at specific locations and time points. In contrast, the input data are primarily from gridded datasets where each pixel represents an average over a defined spatial range and temporal range. This is particularly significant for the ECCO variables, which have the largest spatial grid size of $110\,km$. Consequently, extreme

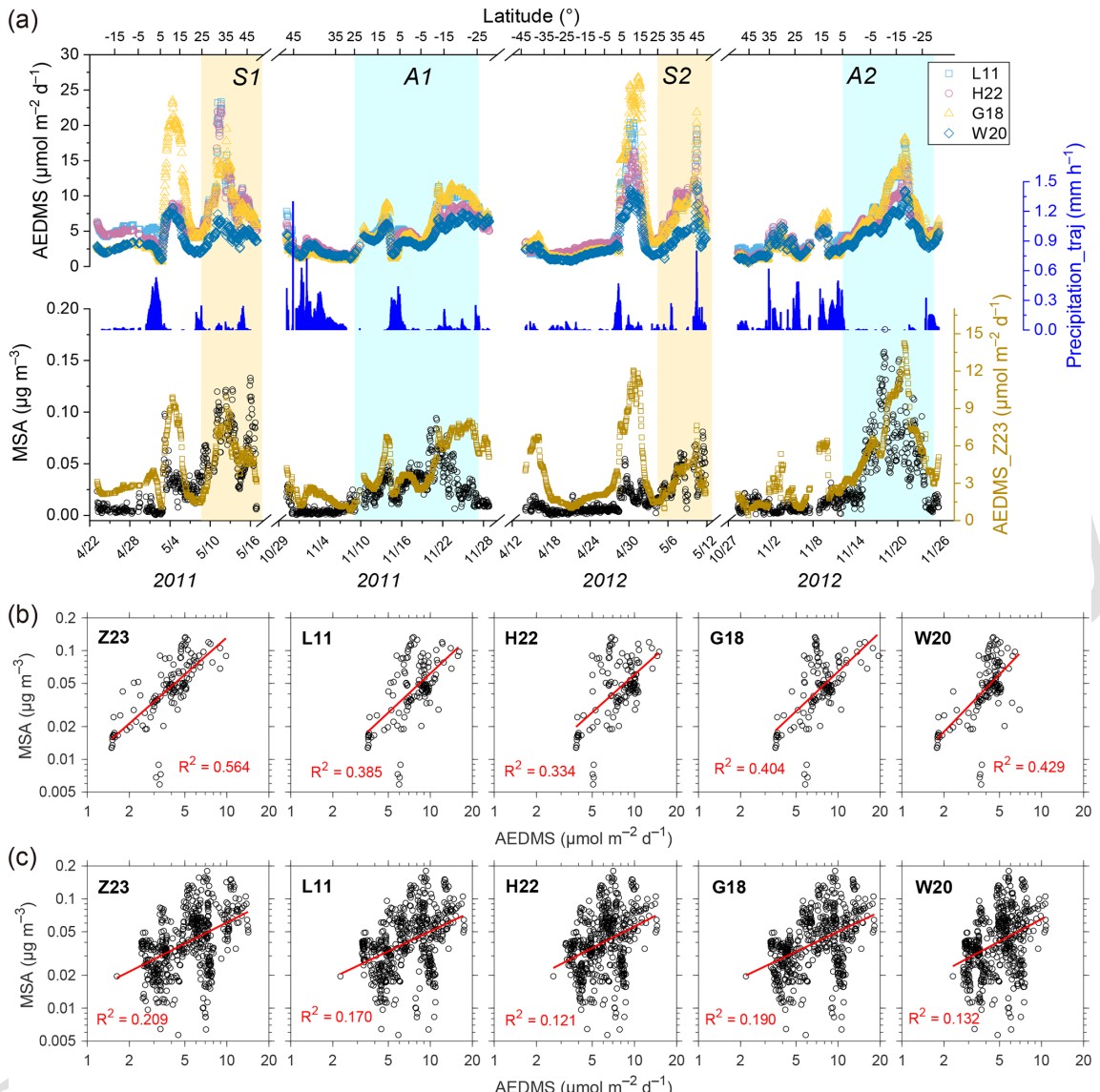

**Figure 13. (a)** Time series of observed MSA concentration, AEDMS calculated based on different DMS concentration datasets, and average precipitation along the backward trajectory (Precipitation_traj) during four Atlantic cruises in 2011–2012. **(b–c)** Correlations between hourly MSA concentration and AEDMS based on different DMS concentration datasets **(b)** during periods S1 and S2 and **(c)** during periods A1 and A2. Data points during periods in which the air mass was within the boundary layer for less than 90 % of the time or Precipitation_traj larger than 0.05 mm h$^{-1}$ were removed.

values at specific locations cannot be accurately captured by the regional averages, resulting in dampened variations among the samples. Secondly, the input data from different sources and the observed sea surface DMS concentrations inherently possess certain uncertainties, which can introduce noise into the ANN learning process. Thirdly, the ANN itself may not be powerful enough to fully capture the complex input–output relationships across different oceanic regions, especially when the samples are scarce under specific environmental conditions. Finally, beyond the nine variables incorporated in this study, other environmental parameters

such as pH (Six et al., 2013; Hopkins et al., 2010) and trace metal elements (Li et al., 2021) can also influence the DMS concentration. Not incorporating these factors may introduce additional biases.

The overall bias for $\log_{10}(\text{DMS})$ is at a similar level at the high- and low-concentration ends, but the DMS concentration on a linear scale is more underestimated in the high-concentration regime than overestimated in the low-concentration regime. As a result, our simulation results may tend to underestimate the annual average DMS concentration and flux. To mitigate this critical bias and reduce model un-

certainty, high-quality input datasets with finer spatial resolution are needed in the future. The high-time-resolution nature of the resulting daily DMS data product would be more valuable if it was accompanied by higher spatial resolution. Expanding the data volume is also crucial for improving model performance. Although the current DMS observational data cover all major oceanic basins, certain regions such as Trades_Pacific remain underrepresented. Advances in online measurement technologies offer promising avenues for acquiring more extensive and convenient observational data (Hulswar et al., 2022). Additionally, incorporating more input features into the model would be beneficial. This necessitates a comprehensive understanding of the spatiotemporal distributions of those input features, and further field measurements are important to this end. Moreover, integrating DMS biogeochemical mechanisms with the machine learning technique, i.e., a hybrid model coupling physical processes with a data-driven approach, may further improve prediction accuracy, generalization, and interpretability (Reichstein et al., 2019).

When using our newly developed DMS dataset, there are two issues that need to be noted. Firstly, there is a significant portion of missing satellite Chl $a$ data during winter in polar regions. In such instances, the modeling data from the CMEMS global biogeochemical multi-year hindcast were used, which may introduce higher uncertainty. We have provided flags indicating the source of the Chl $a$ data for each grid in the dataset. Nevertheless, given the low phytoplankton biomass and extensive sea ice coverage during winter, DMS emissions are typically at the lowest level of the year, so the missing satellite data have a relatively small impact when investigating the subsequent effects of DMS emission on the atmospheric environment. Secondly, since the ANN ensemble model exhibits a limited capacity to accurately reproduce extremely high concentrations of DMS, the DMS concentrations in certain nearshore areas with intensive biological activity may be greatly underestimated.

## 5   Code and data availability

The generated gridded datasets of DMS concentration, total transfer velocity, and flux have been deposited at https://doi.org/10.5281/zenodo.11879900 (Zhou et al., 2024) and can be downloaded publicly. The ANN model code and the MATLAB scripts for data analysis are available from https://doi.org/10.5281/zenodo.12398985 (Zhou, 2024).

## 6   Conclusions

Based on the global sea surface DMS observations and associated data on nine relevant environmental variables, an ANN ensemble model was trained. The ANN model effectively captured the variability of DMS concentrations and demonstrated good simulation accuracy. Leveraging this ANN model, a global sea surface DMS gridded dataset spanning 20 years (1998–2017) with daily resolution was constructed. The global annual average concentration was $\sim 1.71$ nM, which falls within the range of previous estimates, and the annual total emissions were $\sim 17.2$ Tg S yr$^{-1}$. High DMS concentrations and fluxes occurred during summer in the North Pacific (40–60° N), the North Atlantic (50–80° N), the annular band around 40° S, and the Southern Ocean. With this newly developed dataset, the day-to-day changes and interannual variations can be investigated. The global annual average concentration shows a mild decreasing trend ($\sim 0.0035$ nM yr$^{-1}$), while the total emissions remain stable. There were more significant decadal changes in certain regions. Specifically, the annual DMS emissions in the South Pacific and North Pacific showed opposite trends.

To further validate the robustness and advantages of our new dataset, an approach based on air mass trajectories was applied to link the DMS flux and atmospheric MSA concentration. Compared to previous monthly climatologies, the exposure of the air mass to DMS calculated using our new dataset explains a greater amount of the variance in atmospheric MSA concentration over the Atlantic Ocean. Therefore, despite the presence of uncertainties and limitations, the new dataset holds the potential to serve as an improved DMS emission inventory for atmospheric models and to enhance the simulation of DMS-induced aerosols and their associated climatic effects.

## Appendix A: Abbreviations

| AEDMS | Air mass exposure to DMS emission |
|-------|-----------------------------------|
| ANN | Artificial neural network |
| BLH | Boundary layer height |
| CCN | Cloud condensation nuclei |
| Chl *a* | Chlorophyll *a* |
| DMS | Dimethyl sulfide |
| DMSP | Dimethylsulfoniopropionate |
| DO | Dissolved oxygen |
| DSWF | Downward shortwave radiation flux |
| ECCO | Estimating the Circulation and Climate of the Ocean |
| GSSD | Global Surface Seawater DMS (database) |
| Kt | Total transfer velocity |
| MLD | Mixed-layer depth |
| MB | Mean bias |
| MSA | Methanesulfonic acid |
| MSE | Mean square error |
| NAAMES | North Atlantic Aerosols and Marine Ecosystems Study |
| NMB | Normalized mean bias |
| RMSE | Rooted mean square error |
| SI | Sea ice fraction |
| SST | Sea surface temperature |
| SSS | Sea surface salinity |
| WS | Wind speed |

## Appendix B: The weighted resampling strategy

Apart from the data imbalance between coastal and non-coastal regions, an imbalance exists across different DMS concentration ranges. The majority (78.6 %) of DMS concentrations fall within the range of 0.8 to 10 nM ($\log_{10}$(DMS) between $-0.1$ and 1). Samples with DMS concentrations exceeding 15 nM or falling below 0.3 nM only represent 6.9 % of the entire sample set. A weighted resampling strategy was applied to mitigate this imbalance (Fig. S7). We randomly sampled 50 000 samples with replacement from the original sample set. The probability of each sample being selected is proportional to the weighting factor shown as the dashed red line in Fig. S7b, which is dependent on its DMS concentration. First, the probability distribution of initial $\log_{10}$(DMS) values was fitted with a gamma distribution, which is given below and displayed as the blue line in Fig. S7b:

$$f(x) = \frac{1}{\Gamma(k)\theta^k}(x+4)^{k-1}e^{-(x+4)/\theta}. \tag{B1}$$

Here, $k$ and $\theta$ represent the shape parameter and scale parameter: in this case, 100.7 and 0.044, respectively. $x$ is the $\log_{10}$(DMS) value. Since a gamma distribution only takes positive values, we added 4 to the original $x$ used as the dependent variable for distribution fitting. We then obtained a new gamma distribution function with the same mode but a lower shape parameter ($k = 40$) and with $\theta = 0.112$.

The reciprocal of the new gamma distribution function was taken as the weighting factor. As a result, samples exhibiting high or low DMS concentrations are more likely to be selected, whereas those with intermediate concentrations are less likely to be selected. We also controlled the $F_{\text{coastal}}$ value of the resampled data, keeping it equal to 9.7 %. The data distribution of DMS concentrations after the resampling process is shown in Fig. S7c. The fraction of samples with DMS concentrations above 15 nM or below 0.3 nM is elevated to 15.0 %. The 50 000 samples were then randomly split into a training set (80 %) and a validation set (20 %). Since there were duplicate samples in the resampled dataset, the random data split was conducted based on the original sample ID before resampling was performed to ensure that there was no sample overlap between the training and validation sets.

## Appendix C: The calculation of air mass exposure to DMS emissions (AEDMS)

Here, the calculation of the AEDMS index was similar to the calculation of air mass exposure to Chl *a* (AEC) in previous studies (Arnold et al., 2010; Park et al., 2018; Zhou et al., 2021). We adopted a similar approach to that presented in Zhou et al. (2021) but replaced the Chl *a* concentration with the DMS flux, as shown in the following equation:

$$\text{AEDMS} = \frac{\sum_{i=0}^{72}\text{DMS flux}_i \cdot e^{-\frac{t_i}{72}} \cdot \frac{600}{\text{BLH}}}{\sum_{i=0}^{72}e^{-\frac{t_i}{72}}}. \tag{C1}$$

Here, $i$ represents the $i$th trajectory point of the 72 h backward trajectory (the receptor point is the zeroth point). DMS flux$_i$ represents the DMS flux of the pixel in which the $i$th trajectory point is located. DMS flux$_i$ is set to zero if the point is located on land or the air mass pressure is below 850 hPa (usually in the free troposphere with little influence of surface emissions). $t_i$ is the tracking time of the trajectory point (unit: hours) and $e^{-\frac{t_i}{72}}$ is the weighting factor used to assign higher values to regions closer to the receptor point. To better connect with the atmospheric concentrations in the marine boundary layer, normalization by the boundary layer height (BLH) is achieved by including the $\frac{600}{\text{BLH}}$ term. A BLH below 50 m is replaced with 50 m.

**Supplement.** The supplement related to this article is available online at: https://doi.org/10.5194/essd-16-1-2024-supplement.

**Author contributions.** SZ and YC designed the research. SZ, FW, ZX, and KY collected the data and did the data preprocessing. SZ implemented the model development and performed the simulation with assistance from GY, HZ, and YZ. SH, HH, AW, and LP provided the measurement data on atmospheric MSA over the Atlantic Ocean. SZ conducted the data analysis and visualization with advice from YC and XG. SZ and YC wrote the manuscript with inputs from all authors.

**Competing interests.** The contact author has declared that none of the authors has any competing interests.

**Disclaimer.** Publisher's note: Copernicus Publications remains neutral with regard to jurisdictional claims made in the text, published maps, institutional affiliations, or any other geographical representation in this paper. While Copernicus Publications makes every effort to include appropriate place names, the final responsibility lies with the authors.

**Acknowledgements.** We greatly thank the National Oceanic and Atmospheric Administration's Pacific Marine Environmental Laboratory for maintaining the Global Surface Seawater DMS database. We acknowledge Martin Johnson for sharing the code for DMS transfer velocity calculation. We also thank Rich Pawlowicz for developing and sharing the M_Map toolbox for MATLAB (https: //www.eoas.ubc.ca/~rich/map.html, last access: 8 August 2018), which was used in the mapping of this study. Xianda Gong was supported by the Research Center for Industries of the Future (RCIF) at Westlake University and Westlake University Education Foundation.

**Financial support.** This research has been supported by the Natural Science Foundation of Shanghai Municipality (grant no. 22ZR1403800), the National Key Research and Development Program of China (grant no. 2016YFA0601304), and the National Natural Science Foundation of China (grant no. 41775145).

**Review statement.** This paper was edited by François G. Schmitt and reviewed by Murat Aydin and one anonymous referee.

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

## Remarks from the language copy-editor

CE1   Please give an explanation of why this needs to be changed. We have to ask the handling editor for approval. Thanks.

CE2   Please note the insertion. If you would like to edit the unit in the figures, please resubmit the figures with the changes.

## Remarks from the typesetter

TS1   Please confirm.