# Peer review of "A 20-year (1998–2017) global sea surface dimethyl sulfide gridded dataset with daily resolution"

_Earth System Science Data, 2023_

## Author Comment (AC1)

We sincerely thank you for providing very insightful evaluations and suggestions for the manuscript. Our responses to the comments are listed below.

**Color Code:** Referee's comments, Authors response, Proposed changes in manuscript

Line numbers before and inside the bracket refer to those in revised manuscript with and without track of changes, respectively.

The article of Zhou and colleagues presents a novel global gridded dataset of sea-surface DMS concentration and emission based on the ANN technique. Given that DMS is the main biogenic source of atmospheric sulfur globally, the development of approaches that enable the production of detailed DMS emission maps is crucial for atmospheric chemistry and climate studies. The advantage of the new dataset over previous ones is to be found in its daily temporal resolution and multiyear coverage, which (unfortunately) is not matched by increased spatial resolution.

The article is generally well written and gives compelling arguments (e.g. in a strong Introduction) for the wide use of this novel dataset. Beyond the time-resolved fields, other welcome innovations with respect to previous machine learning approaches are the exclusion of time and coordinates as predictor variables (which should enhance model generality and decrease the risk of overfitting) and the validation against fully independent datasets. Below I make some suggestions. I also propose some non-exhaustive corrections to the writing, and encourage the authors to undertake a general check of English grammar.

Specific comments:

L65: please consider citing:

Galí, M., & Simó, R. (2015). A meta-analysis of oceanic DMS and DMSP cycling processes: Disentangling the summer paradox. *Global Biogeochemical Cycles*, *29*(4), 496-515.

Hopkins, F. E., Archer, S. D., Bell, T. G., Suntharalingam, P., & Todd, J. D. (2023). The biogeochemistry of marine dimethylsulfid. *Nature Reviews Earth & Environment*, *4*(6), 361-376.

Thank you for your suggestion. We have added these two references.

L89: the reference to Galí 2021 is incorrect (no machine learning used in that study). The following references to machine learning studies should be included:

McNabb, B. J., & Tortell, P. D. (2022). Improved prediction of dimethyl sulfide (DMS) distributions in the northeast subarctic Pacific using machine-learning algorithms. *Biogeosciences*, *19*(6), 1705-1721.

McNabb, B. J., & Tortell, P. D. (2023). Oceanographic controls on Southern Ocean dimethyl sulfide distributions revealed by machine learning algorithms. *Limnology and Oceanography*, *68*(3), 616-630.

Thank you for pointing out this mistake arising when organizing the reference list. The references to Arctic Ocean should be as following and we have corrected it. The citations you recommended have also been added.

> *Humphries, G. R. W., Deal, C. J., Elliott, S. & Huettmann, F. Spatial predictions of sea surface dimethylsulfide concentrations in the high arctic. Biogeochemistry 110, 287-301 (2012)*

*Qu, B., Gabric, A. J., Zeng, M. & Lu, Z. Dimethylsulfide model calibration in the Barents Sea using a genetic algorithm and neural network. Environ. Chem. 13, 413-424 (2016)*

**Lines 91-99 (87-90):** The machine learning techniques have also been used to simulate the distribution of DMS in the Arctic (Humphries et al., 2012; Qu et al., 2016), North Atlantic Ocean (Bell et al., 2021; Mansour et al., 2023), Northeast Pacific Ocean (McNabb and Tortell, 2022), Southern Ocean (McNabb and Tortell, 2023), and East Asia (Zhao et al., 2022).

L91, entire paragraph: note that higher temporal resolution would be even more valuable if accompanied by higher spatial resolution. Daily resolution (e.g. satellite data) typically shows (sub)mesoscale patterns that are blurred at 1 degree or after monthly averaging.

Thank you for your insightful comment. We agree that high temporal resolution would be more valuable if accompanied by higher spatial resolution. For this study, the spatial resolution is constrained by the ECCO dataset, in which the largest spatial grid size is 110 km, thus we are not able to achieve higher spatial resolution without interpolation. We believe that improving spatial resolution is a direction that needs to be advanced in the future. A short discussion on this issue has been included in Section 4 (**Uncertainties and limitations**).

**Lines 693-697 (564-569):** In terms of the temporal resolution, our product significantly surpasses previous monthly climatologies. However, the higher temporal resolution would be even more valuable if accompanied by higher spatial resolution. In this work, the spatial resolution is limited by the ECCO dataset, where the largest spatial grid size is 110 km. Therefore, we are not able to achieve higher spatial resolution without interpolation. Enhancing the spatial resolution of DMS fields using high-quality input datasets with finer spatial resolution represents a prospective direction for future research.

L124: Is the information on Lat-Lon-Cap 90 really needed here?

Thanks for pointing this out. We agree this information is unnecessary and have removed it. However, to let the reader get a general idea of how large the grid is, we added the grid size range of LLC-90 in Table 1.

L126: using climatological data (nutrients, O2) to produce daily multiyear datasets is a bit paradoxical (as discussed later)

Yes, we agree that using climatological data to produce daily multiyear datasets is not a perfect approach, which may introduce additional uncertainties. We have updated the data sources of those variables. Specifically, we utilized Copernicus-GlobColour Level-4 dataset for Chl *a* and CMEMS global biogeochemical multi-year hindcast for nutrients and DO. Consequently, all input features now originate from multiyear datasets with daily resolution.

**Table 1.** The data sources and related information of variables used for model development, DMS simulation, and flux calculation

| Variable | Data source | URL | Temporal resolution | Temporal coverage | Spatial grid |
|---|---|---|---|---|---|
| DMS | GSSD database | https://saga.pmel.noaa.gov/dms/ | In-situ | Mar. 1972 – Aug. 2017 | - |
| | Other campaigns integrated in Hulswar et al. (2022) | https://data.mendeley.com/datasets/hyn62spny2/1 | In-situ | Feb. 2000 – Jun. 2016 | - |
| Chl *a* | GSSD database | https://saga.pmel.noaa.gov/dms/ | In-situ | Oct. 1980 – Aug. 2017 | - |
| | Copernicus-GlobColour Level-4 | https://data.marine.copernicus.eu/product/OCEANCOLOUR_GLO_BGC_L4_MY_009_104/description | Daily | Sep. 1997 – present | 0.042°×0.04 |
| | CMEMS global biogeochemical multi-year hindcast (only used for the simulation of DMS concentration in polar regions when satellite Chl *a* is unavailable) | https://data.marine.copernicus.eu/product/GLOBAL_MULTIYEAR_BGC_001_029/description | Daily | Jan. 1993 – present | 0.25°×0.25° |
| SST | NOAA OI SST V2 | https://psl.noaa.gov/data/gridded/data.noaa.oisst.v2.highres.html | Daily | Sep. 1981 – present | 0.25°×0.25° |
| MLD | NASA ECCO V4r4 | https://data.nas.nasa.gov/ecco/data.php?dir=/eccodata/llc_90/ECCOv4/Release4 | Daily | Jan. 1992 – Dec. 2017 | LLC90 (22 – 110 km) |
| DSWF | | | | | |
| SSS | | | | | |
| Nitrate | CMEMS global biogeochemical multi-year hindcast | https://data.marine.copernicus.eu/product/GLOBAL_MULTIYEAR_BGC_001_029/description | Daily | Jan. 1993 – present | 0.25°×0.25° |
| Phosphate | | | | | |
| Silicate | | | | | |
| DO | | | | | |
| WS | NASA ECCO V4r4 | https://data.nas.nasa.gov/ecco/data.php?dir=/eccodata/llc_90/ECCOv4/Release4 | Daily | Jan. 1992 – Dec. 2017 | LLC90 (22 – 110 km) |
| SI | NOAA OI SST V2 | https://psl.noaa.gov/data/gridded/data.noaa.oisst.v2.highres.html | Daily | Sep. 1981 – present | 0.25°×0.25° |

Figure 4: Line P stills shows large interannual variability in late summer (Aug) that is not captured by the ANN. This would be clearer if a linear (not log) y-scale was used, and may deserve some discussion.

Thank you for your comment. The y-axis here has been changed to linear scale. Since each data point of Line P program just represents a single measurement taken at a specific time point within the month, rather than the monthly mean, it is difficult to tell the interannual variability based on these measurements. However, we acknowledge that our model falls short in accurately reproducing the extremely high DMS concentrations. This limitation stems from the sparse availability of samples with extreme DMS concentrations for training. We pointed out this issue in the revised manuscript.

**Lines 375-383 (319-326):** Notably, the model generally underestimates high DMS concentrations during summer, particularly those exceeding 10 nM, consistent with earlier discussions. Aggregating data from all campaigns across three regions, the $log_{10}$ space RMSE of simulated DMS concentrations against observations is 0.294, marginally higher than the training set. Most simulated values (87.8%) are within the range of 1/3 to 3 times of observations. The results further evidence that there is no significant overfitting in our model. When data from each campaign are binned, simulations demonstrate high consistency with observations, as depicted in Fig. 6c (RMSE = 0.278, $R^2$ = 0.651). In summary, although our ANN ensemble model may not precisely reproduce small-scale variations and extreme values in specific regions and periods, it reasonably captures overall large-scale variations.

L284: this is also consistent with the phytoplankton spring-summer bloom patterns…

Thank you for your valuable information. We have added it into the discussion.

**Lines 406-410 (341-344):** The other is the subarctic North Atlantic (45°–80° N). A notable increase of DMS concentration starts around 45°–50° N in May and gradually shifts northward beyond 50° N by July (Fig. 7-8). This spatiotemporal evolution pattern corresponds to the evolution of solar radiation intensity and the spring-summer bloom patterns of phytoplankton (Friedland et al., 2018;Yang et al., 2020).

Fig. 6 caption: "for each grid point"

Revised as suggested.

Fig. 8: inclusion of Kt is welcome

L354: are these mean concentrations weighted by pixel (grid cell) area?

Yes, they are also corresponding to area-weighted mean concentrations. We have added this information.

**Lines 495-497 (413-415):** The global area-weighted annual mean DMS concentrations in L11 and H22 are 2.43 nM and 2.26 nM, respectively, which are approximately 41.3% and 31.4% higher than Z23.

L362: this feature of G18 may be due to overestimation of Chl by satellites in coastal regions because of the interference of CDOM and non-algal detrital particles.

Thank you for point this out. We have added it into the discussion.

**Lines 505-508 (421-423):** This characteristic is not fully replicated by other DMS fields, possibly due to the overestimation of Chl *a* by satellites in coastal regions caused by the interference of colored dissolved organic matters and non-algal detrital particles (Aurin and Dierssen, 2012).

Fig. 8, 10, 11: I recommend reporting Kt in m d-1 rather than m s-1

This is a good suggestion. The unit of Kt in these figures and main text have been changed to m d$^{-1}$.

L455: but note that G18 and W20 can be used to produce daily multiyear DMS fields as Z23. This is not possible for interpolated climatologies L11 and H22.

Thank you for your comment. We have added a discussion on it.

**Lines 649-651 (533-535):** It is worth noting that the satellite-based algorithms of G18 and ANN model of W20 can also be utilized to produce daily multiyear DMS fields as Z23. Future investigations could include comparisons with these fields, facilitating a more comprehensive assessment of the performance of each algorithm/model.

Fig. 12a: please use the same colours as in Fig. 9e to distinguish the different algorithms

Revised as suggested.

[Figure]

**Figure 14.** (a) Time series of observed MSA concentration, AEDMS calculated based on different DMS concentration datasets, and average precipitation along the backward trajectory (Precipitation_traj) during four Atlantic cruises in 2011–2012. (b–c) Correlations between hourly MSA concentration and AEDMS based on different DMS concentration datasets (b) during periods S1 + S2 and (c) during periods A1 + A2. Data points during the periods with air mass time fraction within the boundary layer less than 90% or Precipitation_traj larger than 0.05 mm h$^{-1}$ were removed.

Suggested rewording:

L103: "demonstrated" >> "shown, depicted"

L171 "Root(ed) mean square error"

L176 "is larger" >> "exceeds"

L259: "off-line" >> "discrete sampling (Niskin bottle)"

Revised as suggested.

L505: revise grammar

Thank you for pointing this out. We have revised the grammar.

**Line 698 (570):** When using our newly developed DMS dataset, there are two issues that need to be noted.

---

## Author Comment (AC2)

Thank you for your interest in our work and taking the time to review our submission. Your constructive comments and suggestions are of great help to improving our dataset and manuscript. Our responses to the comments are listed below.

**Color Code:** Referee's comments, Authors response, Proposed changes in manuscript

Line numbers before and inside the bracket refer to those in revised manuscript with and without track of changes, respectively.

To fully address reviewers' concerns and further improve the model and subsequent DMS data product, we have reconstructed a new model. This involved incorporating additional training data, changing the data sources of input features, and implementing more reasonable data processing strategies. A new global daily multiyear DMS gridded dataset was obtained, and all figures in the manuscript have been updated accordingly. **Before the point-to-point responses to your comments, we provide an overview of the major modifications we have made in the model development and evaluation.**

1. We included more DMS observation data for training. These data originate from eight campaigns that have not been incorporated into the GSSD database but included in Hulswar et al. (2022). The number of those new samples is 6711.

2. We changed the data sources of Chl $a$, nitrate, phosphate, silicate, and DO. Currently, the time resolutions of all input features are one day.

3. We adjusted the fraction of coastal samples in training, validation, and testing sets to mitigate the overrepresentation of coastal regions. We also applied a weighted resampling strategy in data split process to mitigate the data imbalance between the extreme and moderate DMS values. This treatment will mildly decrease the overall performance (a slight increase in overall RMSE), but significantly reduce the prediction biases for extremely high and extremely low DMS concentrations.

4. Other minor adjustments to make the model development and evaluation procedures more reasonable:

   a. We adjusted the model structure and applied L2 regularization to prevent overfitting.

   b. The fraction of the testing set was elevated, and the figures (e.g., Fig. 4c in the revised manuscript) to demonstrate model performance are based on the testing set, not based on training and validation sets as before.

   c. The discussion of fitting residual and bias has been added.

   d. To further increase the data volume for training, the data of the first two NAAMES campaigns were moved to the training set. We keep the third NAAMES campaign for independent testing.

   e. When comparing the predictions and observations for TRANSBROM SONNE and NAAMES campaigns, the data were not binned into 1°×1° first. Instead, they were binned into 0.05°×0.05°, following the treatment for the training set.

The manuscript by Zhou et al. offers a 20-year (1998-2017) global sea surface dimethyl sulfide (DMS) dataset with daily resolution. The new dataset is developed with an artificial neural network (ANN) ensemble model based on 9 environmental parameters. DMS is produced biogenically in the ocean and its emissions contribute to aerosol radiative forcing in the troposphere. There are a few other global ocean DMS emissions datasets, including one based on an artificial neural network. What makes this dataset unique is that it offers a high time resolution data product covering a 20 year period. The authors claim it is an improved emission inventory of oceanic DMS which can facilitate improved simulations of aerosols derived from DMS. This is a useful dataset with unique features that suits the Earth System Science Data Journal goals of publishing articles on original datasets. However, I would like to see the authors address my comments and questions below before the publication of the manuscript.

My primarily concerns are centered around the comparisons between the ANN product and actual data displayed in Fig. 3. The statistical metrics chosen for arguing good agreement between the ANN product and the observations are R2 and root mean square error (RMSE). These metrics are appropriate for testing the predictive capability of linear regressions, in other words the accuracy of a linear model, but they do not necessarily address the fidelity of the model data to reality. If we are to prefer the ANN data product over the observations to estimate DMS flux, the manuscript needs to present convincing evidence that the model vs. observations relationship is not only linear but also has a slope that centers around 1:1. In this context, it is not enough to show that there is a strong linear relationship between the ANN product and the observations, rather the slope of the linear relationship should be quantified and ideally shown to be statistically indistinguishable from 1.

Thank you for your comments. First, we think that the observational data would absolutely be more reliable than the ANN data product to estimate DMS flux if they were available. However, the DMS observations are still very sparse in both spatial and temporal dimensions, not allowing for analysis of the long-term variability of regional sea surface DMS distribution or investigation of the atmospheric effects of DMS emissions on a regional scale. Therefore, we need to fill in the observational gaps and reconstruct a spatiotemporally continuous sea surface DMS field. Here our approach is utilizing machine learning techniques to capture the underlying dependence of sea surface DMS concentration on other environmental variables, for which spatiotemporally continuous data are available. Then we can combine the machine learning model with the spatiotemporally continuous datasets of those environmental variables to reconstruct the spatiotemporally continuous distributions of sea surface DMS.

Second, we think $R^2$ and RMSE are two most commonly used metrics to test the fidelity of a model for a regression task, as demonstrated in previous studies on using machine learning to predict sea surface DMS concentration (Wang et al., 2020; McNabb and Tortell, 2022, 2023). $R^2$ explains how much of the variance in actual data can be reproduced by the predicted data, while RMSE ($RMSE = \sqrt{\frac{\sum_{i=1}^{n}(\hat{y}_i - y_i)^2}{n}}$, where $\hat{y}_i$ and $y_i$ denote the predicted and actual data, respectively) represents the average difference between predicted and actual values. However, we also acknowledge the importance of considering the slope, as it conveys valuable information on the systematic modelling bias. Ideally, the slope would be statistically indistinguishable from 1. In this study, the slopes are statistically lower than 1, attributed to the systematic biases in both extremely high and extremely low ends. The detailed discussion was given in the response to subsequent comments.

Fig. 3 offers only qualitative information about the value of the slopes. I found the data density color scale helpful in trying to estimate what the slope of the best fits to these scatter plots might be, but the slopes should really be quantified in the manuscript. The manuscript includes a passing reference to a potential bias issue with regards to the coastal region. I agree that, if the bias is limited to high concentrations in that region alone, this would not be a big deal. However, looking at Fig. 3, I suspect that the slopes might be different than unity for multiple regions, although I cannot be not certain without seeing a proper analysis. The fact that the entire analysis is in log-log space makes me more worried because small looking deviations in a log-log linear relationship can result in significant biases in actual concentration and flux calculations.

One can think of many different ways to conduct this type of analyses, but I would advise investigating the residuals of the scatter plots in Fig. 3 from the 1:1 slope versus the DMS_obs. Fitting linear regression lines to these residuals-plots would be a good way to test for biases; ideally these slopes should equal zero within uncertainties, meaning the residuals do not have a positive or negative relationship with DMS_obs.

Thank you for your comments and suggestions. First, using log-log space for figure plot and statistics metric calculation is due to the data distribution of observed DMS concentrations. In linear space, the distribution is highly skewed (Fig. C1). This skewness significantly impacts certain statistical metrics ($R^2$, RMSE, slope, etc.) by being disproportionately influenced by a few high concentration values, thus hindering an accurate reflection of the model's predictive accuracy for the majority of the data. Additionally, plotting scatter plots in linear space results in most data points being obscured. Conversely, after logarithmic transformation of DMS concentrations, a more favorable normal distribution is observed (Fig. C1), allowing the statistical metrics and scatter plots to better reflect the characteristics of the main body of the data. In log space, the absolute deviation between predicted and actual values corresponds to the ratio in linear space. For example, a deviation of 0.2 in log space represents a ratio of 10^0.2, which equals 1.58 in linear space.

We conducted a correlation analysis between prediction residuals and observed values. The corresponding slopes of linear fits (denoted as $S$) are provided in the figures. The slope of the linear fit between predicted and observed values equals 1 minus $S$. For both training and testing sets, the $S$ values for the nine regions are all significantly greater than 0 (Fig. C2-C3), ranging from 0.400 to 0.673. This is attributed to a systematic bias in the model predictions for extremely high and extremely low DMS concentrations, i.e., significant underestimation for higher concentrations and overestimation for lower concentrations. This phenomenon arises from the fact that most DMS observations cluster around intermediate values, with relatively few samples at the extremes (samples with DMS concentrations exceeding 15 nM or falling below 0.3 nM only represent 6.9%). The underestimation of few extremely high values results in a negative mean bias in DMS concentrations in linear space. We calculated the mean bias (MB) of predicted values relative to observed values (linear space) and normalized mean bias (MB/mean observed concentration, denoted as NMB) for each region. The NMB values range from -5.9% to -28.5% across different regions, and an overall NMB corresponding to the entire sample set is -23.6% (Table C2). It is worth noting that these biases are compared against historical DMS observations, which were conducted within a very limited geographical area and time periods. Thus, they cannot be interpreted as the actual mean modeling bias for the entire region.

To improve the model's ability to predict extreme concentration values, we implemented a weighted resampling strategy prior to training to increase the proportion of samples with extreme concentration

values on both sides (see Section 2.3 for details). The new model has indeed shown some improvement on this issue, significantly reducing the *S* values and biases for Polar_N, Polar_S, Westerlies_N_Pacific, Westerlies_N_Atlantic, and Westerlies_S regions (Fig. C4-C5 and Table C2). The overall NMB value is reduced to -16.2%. However, the phenomenon of underestimation for high values and overestimation for low values still persists, and this issue cannot be completely resolved solely through resampling without significantly sacrificing the model's accuracy for intermediate concentrations and avoiding overfitting. Nonetheless, due to the low proportion of extreme values, most residuals are distributed around 0, indicating that the overall performance of the model remains relatively robust. In other words, the general large-scale spatiotemporal variations can be reasonably captured by our model.

We have revised the manuscript and added the discussion accordingly.

[Figure]

**Figure C1.** The probability density function of observed DMS concentrations and those after log transformation.

[Figure]

**Figure C2.** Scatter density plot for prediction residuals of $\log_{10}$(DMS) versus observed values in different regions based on the previous version of our ANN model. The results correspond to training set.

[Figure]

**Figure C3.** Correlations between prediction residuals of $\log_{10}$(DMS) and observed values across different regions based on the previous version of our ANN model. The results correspond to testing set.

**Table C1.** Mean bias and normalized mean bias of predicted DMS concentrations versus observed values across different regions based on the previous version of our ANN model.

| Region | Mean bias (nM) | Normalized mean bias |
|---|---|---|
| Coastal | -1.37 | -28.5% |
| Polar_N | -1.14 | -26.0% |
| Polar_S | -2.09 | -27.4% |
| Westerlies_N_Pacific | -0.89 | -18.8% |
| Westerlies_N_Atlantic | -0.68 | -21.6% |
| Westerlies_S | -0.52 | -20.0% |
| Trades_Pacific | -0.14 | -5.9% |
| Trades_Indian | -0.61 | -21.3% |
| Trades_Atlantic | -0.17 | -8.8% |
| **Global** | **-1.09** | **-23.6%** |

[Figure]

**Figure C4 (Figure S6 in revised manuscript).** Scatter density plot for prediction residuals of $\log_{10}$(DMS) versus observed values in different regions based on the retrained ANN model. The results correspond to training set.

[Figure]

**Figure C5 (Figure 5 in revised manuscript).** Scatter density plot for prediction residuals of $\log_{10}$(DMS versus observed values in different regions based on the retrained ANN model. The results correspond to testing set.

**Table C2 (Table 2 in revised manuscript).** Mean bias and normalized mean bias of predicted DMS concentrations versus observed values across different regions based on the retrained ANN model.

[revised manuscript text omitted]

For example, Figs. 4b,c display linear fits to the data and these slopes are different from unity. This is also noticeable in Fig. 4a as most of the higher concentrations during July-Aug are underestimated by the model and conversely the lower concentrations in the winter tend to be overestimated, leading to a

damped seasonal cycle. I grant that the differences look small in Fig. 4a, but given that this figure too is on a logarithmic scale, it would be good to see a formal quantification of fluxes generated with observed data versus the simulated ones. Do under and over estimations at either end cancel each other out or does one win out over the other, leading to biases in the annual fluxes?

Thanks for your comments on this issue. We have changed the y-aixs of Fig. 6a (Fig. 4a in previous version) to linear scale. We acknowledge that the predictions generally underestimate the higher concentrations during summer, which aligns with our above statement that our ANN model cannot reproduce the extremes perfectly. We have emphasized this point in the revised manuscript. The significant overestimation of the lower concentrations during winter has been largely mitigated with current model.

Our product is a global-scale dataset with a spatial resolution of $1° \times 1°$, primarily focusing on capturing large-scale spatiotemporal variations. Consequently, while we recognize the presence of substantial bias in extreme values within specific small regions, it remains within acceptable bounds for our main objectives. Regarding the quantification of annual fluxes, we think it is difficult to tell the bias in the annual fluxes based on these data, since the observations were only conducted in very limited area and time points. Each single data point represents a single measurement taken at a specific time point within that month rather than a monthly mean. Therefore, we decided not to take further explore on this issue.

**Lines 371-383 (316-326):** It can be seen that the model effectively captures the seasonal variation in Northeast Pacific, which is generally August > June > February (Fig. 6a). However, the small-scale spatial variations can only be partially reproduced by the model in certain campaigns, such as those in June and August of 2007, June of 2009, August of 2012, and August of 2016. Notably, the model generally underestimates high DMS concentrations during summer, particularly those exceeding 10 nM, consistent with earlier discussions. Aggregating data from all campaigns across three regions, the $\log_{10}$ space RMSE of simulated DMS concentrations against observations is 0.294, marginally higher than the training set. Most simulated values (87.8%) are within the range of 1/3 to 3 times of observations. The results further evidence that there is no significant overfitting in our model. When data from each campaign are binned, simulations demonstrate high consistency with observations, as depicted in Fig. 6c (RMSE = 0.278, $R^2$ = 0.651). In summary, although our ANN ensemble model may not precisely reproduce small-scale variations and extreme values in specific regions and periods, it reasonably captures overall large-scale variations.

[Figure]

**Figure 6.** Comparisons between the ANN predictions and observations from fully independent campaigns. (a) Time series of simulation results and DMS observational data obtained from *Line P Program*. The different markers represent different stations of *Line P*. The blue shades cover the data obtained from the cruises included in the GSSD database but with a different method. (b) Scatter plot of simulated versus observed DMS concentrations. (c) The same as panel b but for averaged data of each cruise. The yellow lines and shaded bands are linear fittings and corresponding 95% confidence intervals for $\log_{10}$ space data. The $R^2$ and RMSE displayed in the figure also correspond to $\log_{10}$ space data.

I have a cautionary note when conducting linear fits. For your raw data, I'm guessing the errors for individual DMS_obs will be very small compared with the dynamic range of the dataset, therefore the x errors can be ignored. Likewise, it is probably reasonable to assume y errors are uniform, meaning standard (least-squares in y direction) linear regression analyses could be safely implemented. For the regionally-averaged data show in Fig. 4c, the x errors look very large and both x and y errors look nonuniform, meaning a standard linear regression approach will yield inaccurate estimates of the slope and its confidence band.

Thank you for pointing out that. Actually, the error bar in previous figure does not represents error but the standard deviation of raw data in calculating the average DMS concentration of each campaign. We find it is unnecessary and have removed it.

Some other shorter general comments and questions:

I'm confused about how the data from different time periods are treated during the training and validation steps of ANN model development. As far as I can tell, you use all data from all periods in training and validation. Once you have the ANN model, you input time variable parameters to estimate temporal changes in concentrations and fluxes, is this correct? Your criticism of previous work for using data from

different time periods to estimate a global average flux does not seem justified because you seem to develop your model in the same fashion, or am I missing something?

Thank you for your comments. In this study, the data of each input feature are from global-scale multi-year continuous datasets. For any given sample, if all input features possess valid values, it will be used for training and validation. Those valid samples are mainly distributed between 1997 and 2017. After constructing the model, wherein the dependence of DMS concentration on input features is captured, we use these multi-year continuous input datasets to obtain the global-scale multi-year continuous distribution of DMS. Therefore, we can obtain the interannual variations of sea surface DMS concentrations. Our concern is that previous DMS concentration fields based on interpolation/extrapolation approach combine observations within the same region and month but across different years for spatial interpolation and extrapolation. As a result, the obtained DMS distribution fields are monthly climatological averages rather than multi-year continuous data products, thereby lacking information on interannual variations. We have made a modification to the sentence to make it clearer.

Lines 89-90 (80-81): Furthermore, the observational data from different years within a particular month were combined together for interpolation and extrapolation, and the interannual variations cannot be investigated by this approach.

It would be good see how much data each region contributes to the full dataset. The coastal region appears to contribute the most even though the emissions from the coastal regions constitute only 3% of the global DMS flux, and conversely the trades regions have little data even though the integrated fluxes in these regions are high. Did you try training the model without the coastal data to see if the model results change?

Thank you for your comments. The number of data points in each region (the value of n) has been provided in the scattering plots (Fig. 3 in previous manuscript, and Fig. 4, S4 and S5 in revised version). As you pointed out, there is a notable disparity in data density between coastal and open ocean regions. Specifically, samples from the Coastal biome constitute 27.3% of the entire sample set, despite the coastal area representing only 9.7% of the global ocean area. This disproportionality could potentially lead to a bias in the model, emphasizing coastal regions and decreasing the capability of capturing the data patterns in open oceans. To mitigate this data imbalance issue, we adjusted the data distribution during model training and validation processes. Specifically, we adjusted the fraction of coastal samples to match the area fraction. The details are explained in the manuscript.

Lines 194-200 (169-175): It is noteworthy that there are 11,237 samples in the Coastal region, constituting 27.3% of the entire sample set, despite the Coastal biome accounting for only 9.7% of the global ocean area. Given the distinct seawater physiochemical and biological conditions in coastal seas compared to other regions, the disproportionately higher density of samples within the Coastal biome might cause the model to overly prioritize this region. To mitigate this data imbalance and ensure the model captures broader patterns in open oceans, we adjusted the data distribution during model training and validation processes. Specifically, we adjusted the fraction of coastal samples to match the area fraction. Further details are provided in the subsequent section and visualized in Fig. 3a.

Lines 206-225 (181-199): We randomly selected 10% of the samples (n = 4,116) to be entirely excluded from training, as a testing subset for global validation and overfitting test. The testing subset was

controlled to contain a proportion of coastal samples (denoted as $F_{coastal}$) at 9.7%. Specifically, 401 samples were randomly selected from Coastal biome, while 3,715 samples were selected from other biomes to compose the testing subset. Then, the remaining samples (n = 37,041) were utilized for training and cross validation. Apart from the data imbalance between coastal and non-coastal regions, there exists an imbalance across different DMS concentration ranges. As shown in Fig. 3b, the majority of DMS concentration values (78.6%) fall within the range of 0.8 to 10 nM ($\log_{10}$(DMS) between -0.1 to 1). Samples with DMS concentrations exceeding 15 nM or falling below 0.3 nM only represent 6.9% of the entire sample set. Here we implemented a weighted resampling strategy to mitigate this imbalance and enhance the model's capability in predicting extreme values. We randomly sampled 50,000 samples with replacement from the remaining sample set. The probability of each sample being selected is proportional to the weighting factor shown in Fig. 3b, which is dependent on its DMS concentration. Samples exhibiting high or low DMS concentration values are more likely to be selected, whereas those with intermediate concentrations are less likely to be selected. The details of the weighting factor are explained in Appendix B. We also controlled the $F_{coastal}$ value of the resampled data equals to 9.7% by the same method as described above, i.e., applying the resampling process to coastal and non-coastal samples separately and combining them together afterwards. The data distribution of DMS concentrations after the resampling process is shown in Fig. 3c. The fraction of samples with DMS concentrations above 15 nM or below 0.3 nM is elevated to 15.0%. The 50,000 samples were then randomly split to a training set (80%) and a validation set (20%). Since there are duplicate samples in the resampled dataset, the random data split was conducted based on the original sample ID before resampling to ensure that there was no sample overlap between the training and validation sets.

What are the contributions of the 9 different model parameters to the final outcome? Which parameters carry more important information according to your ANN model?

We have done an extensive analysis of the feature importance for each region, and the results are depicted in Fig. C6. Chl $a$ is the most important factor in coastal region and the second most important factor in Arctic region. This aligns with the existing understanding that the oceanic DMS cycling in these regions is under a "bloom-forced regime" (Toole and Siegel, 2004), wherein the DMS production is controlled by the phytoplankton biomass. The importance of SSS in Arctic region may reflect the inflow of more saline, warmer and nutrient-rich Atlantic water mass (the so-called "Atlantification"), where high abundances of *Phaeocystis pouchetii* were generally found (Vogt et al., 2012; Schoemann et al., 2005). In other regions, DSWF and MLD generally rank among the top three key factors, consistent with the significant role of solar radiation dose in controlling the DMS variation in upper mixed layer (Vallina and Simó, 2007; Vallina et al., 2007). Macronutrients and SST also demonstrate noteworthy importance in specific regions. However, we think delving into the explanation of the model will make the manuscript too long and it is not the main objective of the manuscript. Hence, we decided not to include it.

[Figure]

**Figure C6.** The feature importance in each region.

Was the ANN allowed to freely chose model equations, did you impose any restrictions or try other models?

Thanks for raising these questions. In artificial neural networks (ANN), the model's architecture is inherently characterized by a network of connections (weighting factors) linking nodes across adjacent layers, augmented by a bias term in each hidden layer node after aggregating preceding layer information, and culminating in an activation function. Therefore, an ANN cannot be expressed by an explicit mathematical equation.

Throughout the training process, we implemented L2 regularization to counteract overfitting and improve the model generalization. Different lambda values of L2 regularization spanning 2E-4, 5E-4, 1E-3, and 5E-3 have been tried. We have also employed early stopping to further mitigate overfitting. This mechanism halts the training process if the validation loss exceeds or equals the minimum validation loss computed so far 20 times in a row. We have also tried other machine learning algorithms, including random forest and gaussian processing regression. As for ANN, we investigated different architectures, including single-layer configurations with 10, 20, and 30 nodes, as well as two-layer configurations with 10, 15, and 20 nodes in each layer. By evaluating the model performance on testing set, we determined that a two-layer ANN with 15 nodes in each layer yielded optimal performance. We added more detailed information on the current model into the manuscript. However, the detailed information of how we determine the model structure is not included, since we believe it is not of significance for the main objective of this study and the manuscript.

**Lines 226-230 (200-204):** Our feedforward fully connected neural network comprises two hidden layers, with 15 nodes in each layer. The activation functions for the first and second layers are ReLU and tanh, respectively. We applied L2 regularization (lambda = 5E-4) to counteract overfitting. The loss function is mean square error (MSE). Training stops if the validation loss is greater than or equal to the minimum validation loss computed so far 20 times in a row. The training processes were carried out with Statistics and Machine Learning Toolbox on Matlab 2022b.

Minor comments/corrections as they appear in the manuscript:

Line 81-82: This sentence here gives the impression that you are not using all data from all years with equal weight.

Thank you for your question. We are using all samples with valid input feature data for model development. Please see our response to the first shorter general comment.

Line 113: Are you using exactly the same data that went into Hulswar et al (2022)?

For the previous version of the model, we relied on the GSSD database for its development. For the current version, we have incorporated additional data that were not encompassed within the GSSD database but were utilized in Hulswar et al (2022). Because there are three input features (SST, MLD, and SSS) sourced from NASA ECCO dataset, which just extends up to the year 2017, the data after 2017 in Hulswar et al (2022) were not integrated.

**Lines 125-127 (115-117):** Hulswar et al. (2022) consolidated other DMS measurements not included in the GSSD database to establish an updated DMS climatology. Here we incorporated these additional data predating 2017, originating from 8 campaigns (number of samples = 6,711).

Lines 128-131: What happens in SI covered areas? What level of SI cover lead to zero emissions?

Thank you for your questions. In regions covered by sea ice, the available open-water surface area for sea-to-air gas exchange is diminished. Thus, to calculate regional DMS flux, we apply a direct scaling approach the original total transfer velocity by the open-water surface area fraction. This fraction is represented by 1 minus the sea-ice fraction. In this calculation, a sea-ice fraction of 1 will lead to zero emissions. The details are given in Section 2.4.2. We have added a reference to that section in this sentence.

**Lines 144-146 (130-132):** The surface wind speed (WS) and sea ice fraction (SI) data are also needed in the calculation of sea-to-air flux (details are provided in Section 2.4.2). Here we utilized the daily 10-meter WS data from ECCO V4r4 and the daily SI data from NOAA OI SST V2.

**Lines 271-278 (227-232):** The sea-to-air fluxes of DMS were calculated on the basis of simulated surface DMS concentrations following equation (1):

$$DMS\ flux = Kt \times (DMS_w - \frac{DMS_a}{H}) \tag{1}$$

Here $DMS_w$ and $DMS_a$ are DMS concentrations in surface seawater and air, respectively. $H$ is Henry's law constant of DMS. Since $\frac{DMS_a}{H}$ is usually $\ll DMS_w$, this term was omitted in the calculation. $Kt$ is the total transfer velocity considering the sea ice coverage fraction ($SI$):

$$Kt = k_t \times (1 - SI) \tag{2}$$

Line 143: Are the SeaWiFS and Aqua-MODIS data in reasonable agreement?

Yes, they agree quite well during their overlapping period, as displayed in Fig. C7. For the current version of the model, we utilized the Copernicus-GlobColour Level-4 dataset, which integrates multiple upstream sensors including SeaWiFS, MODIS-Aqua & Terra, MERIS, VIIRS-SNPP & JPSS1, and OLCI-S3A & S3B, and an interpolation procedure is applied to fill in missing data. This dataset shows good agreement with in-situ Chl $a$ observations.

[Figure]

**Figure C7.** Comparison between Chl $a$ data derived from Aqua-MODIS and SeaWiFS, corresponding to DMS samples collected during the overlapping period of these two sensors.

**Lines 132-136 (122-125):** Chl $a$ data were obtained from both in-situ observations, co-located with DMS data, and satellite remote sensing products (Copernicus-GlobColour, Level-4, daily, 0.042°×0.042°). The Copernicus-GlobColour Level-4 dataset integrates multiple upstream sensors including SeaWiFS, MODIS-Aqua & Terra, MERIS, VIIRS-SNPP & JPSS1, and OLCI-S3A & S3B, with an interpolation procedure applied to fill missing data (Garnesson et al., 2019).

[Figure]

**Figure S2.** The comparison between the in-situ Chl *a* from GSSD database and the Copernicus-GlobColour Level-4 satellite-retrieved Chl *a* data. n is the number of samples. $R^2$ and RMSE correspond to $\log_{10}$ space data.

Fig. 4a: The markers look quite faint on my screen. I suggest sharper colors.

Revised as suggested.

Lines 339-344: Refer to Fig. 9 somewhere.

Thank you for your suggestion. We have added the reference to Fig. 9 (now Fig. 11).

**Lines 478-479 (398-399):** Here we compare the distributions of DMS concentration derived from our ANN simulation (referred to as Z23) with four previously constructed climatologies (Fig. 11),

Lines 388-390: What drives the trends in Kt?

Thank you for bringing this up. In mid- and low-latitudes, both increasing WS and rising SST have a positive effect on Kt, but WS is generally the dominant driver (Land et al., 2014). This point is discernible through the nearly identical spatial pattern of Kt changes corresponding to WS changes, contrasting with the less consistent spatial pattern in the changes of SST. Here we have modified the sentence to point this out.

**Lines 548-550 (449-451):** Due to the primary influences of increasing WS and secondary impact of rising SST in most mid- and low-latitude regions (Fig. S8), the Kt of DMS also showed an overall increasing trend, especially in the eastern Pacific and Atlantic Ocean (Fig. 12b).

---

## Author Response (AR2)

Dear Dr. Murat Aydin,

Thank you very much for reviewing our revised manuscript. Your insightful comments are very helpful in further improving the quality and completeness of our work. Please find our replies below in blue. The proposed changes in manuscript are shown in green. Specified line numbers before and inside the bracket refer to those in revised manuscript with and without track of changes, respectively.

Best regards,

Ying Chen on behalf of all authors

I acknowledge the extensive nature of revisions to the paper. While many of the revisions resulted in improvements, some feel like a step backward. Given that this the second go around, I will not dwell on minor issues. I do recognize the value of the ANN data set in terms of high resolution in both temporal and spatial scales. My main concerns are related to the monthly and annual fluxes, specifically the fidelity of these estimates to reality as defined by the available observations.

There is only one way to test the accuracy of the ANN data product: it has to be compared with the DMS obs that underlie the training of the machine learning process. In my first review, I suspected that the linear regressions between the DMS obs and the ANN estimates yielded slopes significantly different than 1 and suggested the residuals might be correlated with the observations. I further added that the statistical metrics they relied on were insufficient to adequately evaluate the accuracy of the data product. The additional analyses the authors conducted based on the review confirm my suspicions were correct. While I appreciate the effort that went into the revisions, I do have misgivings about a major aspect revision they implemented and suggest further revisions.

The weighing scheme implemented to increase the influence of low and high concentrations on the results is a data analysis gimmick aimed at improving the linear regressions with respect to the deficiencies I outlined in the first review. I do not believe it is appropriate to manipulate the distribution of the training data in this manner unless they are real life reasons (related to the real world ocean and how it has been sampled) why lower and higher DMS concentrations are underrepresented in the observational data sets.

The manuscript offers no such justification. As such, they would be better of presenting the original ANN results as the main data product and offer the weighing-based results as supplementary analysis. When referring to this supplementary analysis, you should discuss in the main body of the manuscript why it was conducted. In my view, the implemented weighing scheme does not make enough of a difference in the end and I remain unconvinced that the problematic aspects of the linear regressions are caused by extreme concentrations that constitute a small fraction of the data set. There appears to be a systematic issue for reasons that remain unclear to this reviewer.

Thanks for your comments on this issue. The weighted resampling strategy or over-sampling of the minority class is a widely used approach in machine learning to deal with data imbalance and improve the model performance and generalization (Haibo et al., 2008; Yu and Zhou, 2021; Chawla et al., 2002). During the ANN training process, the model tends to focus on optimizing the majority class of data, such as samples with moderate DMS concentrations in this study, and may overlook data patterns within the minority class. Increasing the proportion of the minority class in the training process can let the model learn more information from these samples. However, given the limited improvements in this study, we acknowledge that there are other important issues contributing to the systematic bias. Potential reasons include (1) a mismatch in the spatial and temporal scales between the input and target, (2) uncertainties associated with the input data and DMS measurements, (3) limited capability of machine learning model to fully capture the complex input-output relationships, and (4) the effects of other environmental factors not incorporated in this study. The details are discussed in Section 4 (***Uncertainties and limitations***).

Here we have reverted to using the original ANN model construction to generate the DMS data product. However, some updates made in the second version, including the update of input data sources and the inclusion of more DMS observations, are retained. The results after implementing the weighted resampling scheme have been moved to supporting information and regarded as an approach to test whether the systematic bias is attributable to data imbalance. The details of how the weighted resampling scheme was conducted have been moved to Appendix. The simulated DMS distributions does not show significant differences compared with the second version. All figures and values in the manuscript have been updated.

[revised manuscript text omitted]

$$f(x) = \frac{1}{\Gamma(k)\theta^k}(x + 4)^{k-1}e^{-(x+4)/\theta} \tag{A1}$$

Here $k$ and $\theta$ represent the shape parameter and scale parameter, in this case, 100.7 and 0.044, respectively. $x$ is the $\log_{10}$(DMS) value. Since gamma distribution only takes positive values, we added 4 to the original $x$ as the dependent variable for distribution fitting. We then obtained a new gamma distribution function with the same mode but lower shape parameter, in which $k = 40$ and $\theta = 0.112$. The reciprocal of the new gamma distribution function was taken as the weighting factor. As a result, samples exhibiting high or low DMS concentration values are more likely to be selected, whereas those with intermediate concentrations are less likely to be selected. We also controlled the $F_{coastal}$ value of the resampled data equal to 9.7%. The data distribution of DMS concentrations after the resampling process is shown in Fig. S7c. The fraction of samples with DMS concentrations above 15 nM or below 0.3 nM is elevated to 15.0%. The 50,000 samples were then randomly split to a training set (80%) and a validation set (20%). Since there are duplicate samples in the resampled dataset, the random data split was conducted based on the original sample ID before resampling to ensure that there was no sample overlap between the training and validation sets.

Further, I do not like the fact that the comparison of the training data versus the observations are not shown in the main manuscript anymore. If the number of figures in the manuscript is a problem, I suggest moving the residual figures to the supplement and showing the main comparison figures with respect to both the training and test data in the main body. The slope values should be displayed in all sets of figures. Most readers may not readily infer the implications of trending residuals and the manuscript does not offer a detailed enough discussion.

Thank you for your suggestions. We have moved the comparison for training set back to the main text and placed the residual plots in the supporting information. The slopes of linear regression between predictions and observations are also added in the figures. The discussions for the potential reasons of slopes lower than unity are provided in Section 4, as mentioned in the response to the above comment.

[Figure]

**Figure 3.** Comparisons between ANN-simulated and observed DMS concentrations. (a) Scatter density for simulated versus observed DMS concentrations of the samples used in ANN training. (b) Comparison between the simulated versus observed DMS concentrations of testing set. (c) Comparison between the simulated versus observed DMS concentrations of the samples used in ANN training across 9 regions. The number of data points (n), $\log_{10}$ space $R^2$, root mean square error (RMSE), and linear regression slope are also displayed.

[Figure]

**Figure 4.** Comparisons between the simulated versus observed DMS concentrations of the testing set across 9 regions.

A welcome revision to the manuscript is the inclusion of regional mean and normalized mean bias estimates presented in Table 2. However, this is the bare minimum necessary since the positive and negative biases that occur at high and low ends of the concentrations tend to cancel out during the averaging, therefore hindering insight into the biases at grid scale let alone how these biases impact the regional and global fluxes. I'm willing to accept these outstanding issues as subjects of future work as long as they are pointed out in the paper.

Thanks for your comments. We acknowledge that this is a critical issue needs to be addressed in the future. We have pointed out that the negative biases at high end of the concentrations will be partially cancelled out by the positive biases at low end during the averaging and the bias at a specific grid could be much larger. In Section 4, we have also proposed several measures that can be taken in the future to mitigate this bias.

Lines 306-308 (268-269): On the other hand, the negative biases at high end of the concentrations are partially cancelled out by the positive biases at low end during the averaging over the entire region. The bias at a specific grid could be much larger.

Lines 609-622 (553-565): The overall bias for $\log_{10}$DMS is at a similar level between high- and low-concentration ends, but the DMS concentration on a linear scale is more underestimated in the high-concentration regime than it is overestimated in the low-concentration regime. As a result, our simulation results may tend to underestimate the annual average DMS concentration and flux. To mitigate this critical bias and reduce model uncertainty, high-quality input datasets with finer spatial resolution are needed in the future. The high-time resolution nature of the resulted daily DMS data product would be more valuable if accompanied by higher spatial resolution. Expanding the data volume is also crucial for improving model performance. Although the current DMS observational data covers all major oceanic basins, certain regions such as the Trades_Pacific remain underrepresented. Advances in online measurement technologies offer promising avenues for acquiring more extensive and convenient observational data (Hulswar et al., 2022). Additionally, incorporating more input features to the model would be beneficial. This necessitates a comprehensive understanding of the spatiotemporal distributions of those input features, and further field measurements are important to this end. Moreover, integrating DMS biogeochemical mechanisms with machine learning technique, i.e., a hybrid model coupling physical processes with data-driven approach, may further improve prediction accuracy, generalization, and interpretability (Reichstein et al., 2019).